# Using a Surrogate-Assisted Bayesian Framework to Calibrate the Runoff Generation Scheme in E3SM Land Model V1

Donghui Xu[1], Gautam Bisht[1], Khachik Sargsyan[2], Chang Liao[1], L. Ruby Leung[1]

[1]Atmospheric Sciences and Global Change Division, Pacific Northwest National Laboratory, Richland, WA, USA
[2]Sandia National Laboratories, Livermore, CA, United States

*Correspondence to*: Donghui Xu (donghui.xu@pnnl.gov)

**Abstract.** Runoff is a critical component of the terrestrial water cycle and Earth System Models (ESMs) are essential tools to study its spatio-temporal variability. Runoff schemes in ESMs typically include many parameters so model calibration is

10 necessary to improve the accuracy of simulated runoff. However, runoff calibration at global scale is challenging because of the high computational cost and the lack of reliable observational datasets. In this study, we calibrated 11 runoff relevant parameters in the Energy Exascale Earth System Model (E3SM) Land Model (ELM) using a surrogate-assisted Bayesian framework. First, the Polynomial Chaos Expansion machinery with Bayesian Compressed Sensing is used to construct computationally inexpensive surrogate models for ELM-simulated runoff at 0.5° × 0.5° for 1991-2010. Error metric between

15 the ELM simulations and the benchmark data is selected to construct the surrogates, which facilitates efficient calibration and avoids the more conventional, but challenging, construction of high-dimensional surrogates for the ELM simulated runoff. Second, the Sobol index sensitivity analysis is performed using the surrogate models to identify the most sensitive parameters, and our results show that in most regions ELM-simulated runoff is strongly sensitive to 3 of the 11 uncertain parameters. Third, a Bayesian method is used to infer the optimal values of the most sensitive parameters using an observation-based global

runoff dataset as the benchmark. Our results show that model performance is significantly improved with the inferred parameter values. Although the parametric uncertainty of simulated runoff is reduced after the parameter inference, it remains comparable to the multi-model ensemble uncertainty represented by the global hydrological models in ISMIP2a. Additionally, the annual global runoff trend during the simulation period is not well constrained by the inferred parameter values, suggesting the importance of including parametric uncertainty in future runoff projections.

## 1 Introduction

Runoff is an essential source of freshwater resource, and its variability has profound socio-economic impacts (Hall et al., 2014; Vörösmarty et al., 2000). Flooding in wet regions during peak streamflow is among the most impactful natural hazards of all weather-related events in terms of fatalities and material costs (Doocy et al., 2013). However, higher streamflow replenishes reservoirs that help provide water for agriculture and hydropower generation, and transports nutrients to the

30 floodplain. Drought is a form of hydrological extreme that can also result in immense damages to the ecosystem and agriculture

(Mishra and Singh, 2010). It is associated with abnormally low runoff, especially in arid and semi-arid regions. Therefore, understanding the spatial and temporal patterns of runoff is crucial for flood control, water management, crop yield, ecosystem services, etc. The runoff variability has been impacted by human-induced land use and climate change (Milly et al., 2008; Fischer and Knutti, 2016; Bosmans et al., 2017; Dai, 2013; Xu et al., 2021a), and the changes are projected to be more significant towards the end of this century (Xu et al., 2021a).

The spatial and temporal patterns of runoff and its response to climate change for water security assessments and water management are commonly studied using Earth System Models (ESMs) (Milly et al., 2002; Hirabayashi et al., 2013; Schewe et al., 2014). Current generation ESMs have large uncertainty in the simulation of runoff and its changes under future scenarios. However statistical methods have been applied recently to reduce uncertainty in model predictions (Yang et al., 2017; Gosling and Arnell, 2011; Lehner et al., 2019; Xu et al., 2021a). Uncertainties in ESMs simulation of runoff stem from uncertain model inputs, model structural uncertainty, and parametric uncertainty (Sun et al., 2013; Giuntoli et al., 2018). Input uncertainties consist of uncertainties in atmospheric forcing and land surface cover data that can be reduced by improving observation quality as more data become available. Model structural uncertainty is due to knowledge gaps or simplifications of the physical processes of the earth system. Specifically, the typical coarse resolution (~100 km) of ESMs cannot capture a few of the key physical factors that control runoff generation process such as terrain and soil variations. Downscaling methods have been developed to reduce model bias when projecting the changes of hydrological variables from the coarse resolution ESM simulation to a fine resolution (Tebaldi et al., 2005; Knutti et al., 2010; Xu et al., 2019). Recent development in the Energy Exascale Earth System Model (E3SM) has introduced a sub-grid topography based downscaling of precipitation (Tesfa et al., 2020) to understand the role of topography in hydrological processes. Over the past few decades, the land component of ESMs has continuously been improved by developing new representations of physical processes, such as implementing variable soil thickness (Brunke et al., 2016), solving the variably saturated flow in groundwater dynamics (Bisht et al., 2018), including land-river interactions (Decharme et al., 2019; Xu et al., 2021b), representing lateral subsurface flow (Swenson et al., 2019), and increasing spatial resolution (Haarsma et al., 2016). While these advances improve our understanding of the earth system, they may not lead to reduced uncertainties in future projections (Knutti and Sedláček, 2012; Lehner et al., 2020). This is because parametric uncertainty may increase as new processes are included in the model. The uncertainty in ESM simulated runoff must be reduced before reliable conclusions can be drawn regarding ESM projections of future changes in the runoff characteristics.

The parametric uncertainties in simulated runoff can be reduced by model calibration (Gupta et al., 1998). Previous studies have shown that it is possible to constrain the uncertainty of runoff by calibrating the relevant model parameters at regional scale (Ray et al., 2015; Sun et al., 2013; Sheng et al., 2017; Xie et al., 2007; Troy et al., 2008; Hou et al., 2012; Huang et al., 2013). Hou et al. (2012); Huang et al. (2013) identified the most sensitive hydrologic parameters of the Community Land Model (CLM) for simulating runoff and surface energy fluxes at a few selected watersheds and flux tower sites in the US. They found that reducing the dimensionality of uncertain parameters using sensitivity analysis speeds up the calibration processes (Huang et al., 2013). Consequently, Sun et al. (2013) successfully applied a Bayesian inversion approach to estimate

the optimal parameters to improve the performance of runoff generation in CLM. Troy et al. (2008) proposed an efficient framework to calibrate the Variable Infiltration Capacity (VIC) model for the contiguous US by interpolating the calibrated parameters from small gauged basins. While previous studies performed comprehensive model calibration of runoff at regional scales, it remains challenging to calibrate land components of ESM at global scales due to (1) the lack of runoff observations and (2) the high computational cost of running a large ensemble of global land model simulations. For (1), it is common to validate land models with streamflow (i.e., flow rate accumulated from runoff within a drainage area) observation (Li et al., 2015; Krysanova et al., 2020; Beck et al., 2017; Zhang et al., 2016), as runoff is not directly measured. However, routing the runoff to simulate streamflow at coarse resolution introduces additional uncertainties due to the representation of stream network (Wu et al., 2011; Liao et al., 2022) and river channel geometry (Andreadis et al., 2013). A recent observation-based global runoff dataset (GRUN; Ghiggi et al., 2019) provides a good benchmark for calibrating runoff generation related parameters without the needs of coupling the land model with a river routing model. For (2), tens of thousands of simulations are typically needed for parameter calibration when the parameter dimension is high, but it is not computationally feasible to run a large ensemble of ESM simulations at global scale.

The computational cost of model calibration can be significantly reduced by using an uncertainty quantification (UQ) framework that develops surrogate models of complex physical models. UQ frameworks include several steps: 1) Construction of a surrogate model that can mimic the behaviour of a physical model; 2) Identification of sensitive parameters to reduce the dimensionality of uncertain parameters; 3) Use of the parameter inference process to constrain the parametric uncertainty by comparing surrogate model prediction against observation. The surrogate modelling approach has received wide attention in hydrological applications (Razavi et al., 2012; Ivanov et al., 2021; Wang et al., 2014) to calibrate large-scale land models in terms of different hydrological processes (Gong et al., 2015; Lu et al., 2018; Müller et al., 2015; Huang et al., 2016; Ray et al., 2015; Sargsyan et al., 2014; Ricciuto et al., 2018). Multiple methods falling into the class of surrogate models include Gaussian process models, artificial neural networks, support vector machines, and polynomial chaos expansions (PCEs). In this study, we rely on PCEs as convenient machinery for uncertain input parameter representation and surrogate construction. The PCE surrogate captures the complex, non-linear behaviour of the physical model through a learned polynomial expansion. This method also provides convenient global sensitivity analysis (Dwelle et al., 2019). Further, we employ Bayesian compressive sensing (BCS) to arrive at sparse PCEs that include only polynomial terms relevant to the model, thus facilitating PCE surrogate construction in the presence of a large number of uncertain inputs and a relatively small number of model simulations (Sargsyan et al., 2014). Once the surrogate model is constructed, it replaces the expensive physical model in simulation-intensive studies such as global sensitivity analysis and parameter inference.

The objective of this work is to use the UQ framework to improve the performance of runoff generation at monthly scale and quantify the associated parametric uncertainty in the E3SM Land Model version 1 (ELM-v1) (E3SM; Golaz et al., 2019). This study is organized in the following structure. We briefly describe the runoff generation process in ELM-v1, the UQ framework, and the data used in this study in Sections 2, 3, and 4, respectively. In Section 5, we first present the validation of the surrogate models, sensitivity of simulated runoff to the uncertain parameters, dimensional reduction of uncertain

parameters, and estimation of optimal parameters. Then we evaluate the performance of ELM-simulated runoff with the optimal parameters, the runoff sensitivity to precipitation, and the changes due to the use of optimal parameters on ELM-simulated water- and energy-related variables against various benchmarks using the ILAMB package (Collier et al., 2018). Lastly, we present the simulated runoff uncertainty associated with parameters and theirs impacts on runoff trends at global scale. Section 6 discusses the limitations of this work, followed by the conclusions in Section 7.

## 2 E3SM Land Model

### 2.1 Runoff generation scheme in ELM-v1

The ELM-v1(hereafter, v1 is omitted) was developed based on the Community Land Model 4.5 (CLM4.5; Oleson et al., 2013) to understand the water availability and water cycle extremes (Leung et al., 2020). The new physical processes added in ELM to better represent the terrestrial water cycle include a variably saturated flow model (Bisht et al., 2018), a soil erosion model (Tan et al., 2020), dynamic roots (Drewniak, 2019), and a two-way coupled irrigation scheme (Zhou et al., 2020). The runoff generation in ELM is based on the simple TOPMODEL-based runoff parameterization (SIMTOP; Niu et al., 2005) in which the total runoff ($R_{total}$) consists of three components: surface runoff ($R_{over}$, e.g., saturation excess runoff), surface water runoff ($R_{h2osfc}$, e.g., surface water drainage from depressions/wetlands), and subsurface runoff ($R_{drai}$):

$$R_{total} = R_{over} + R_{h2osfc} + R_{drai} \qquad \text{Eq. (1)}$$

A fraction of the flux of water reaching the soil surface ($q_{liq}$) generates surface runoff and the fraction is determined by the saturation fraction ($f_{sat}$) of the grid cell:

$$R_{over} = f_{sat}q_{liq} \qquad \text{Eq. (2)}$$

$$f_{sat} = f_{max}\exp(-0.5f_{over}z_{\nabla}) \qquad \text{Eq. (3)}$$

where $f_{max}$ represents the maximum saturation fraction for a given grid cell that is calculated with high-resolution compound topographic indices, $f_{over}$ is a decay factor, and $z_{\nabla}$ is the water table depth.

ELM includes surface water storage to represent inland/wetland surface water dynamics (Ekici et al., 2019). When the surface water storage is fully filled, surface water runoff is generated:

$$R_{h2osfc} = k_{h2osfc}f_{connected}\left(W_{sfc} - W_c\right)\frac{1}{\Delta t} \qquad \text{Eq. (4)}$$

where $k_{h2osfc}$ represents the linear storage coefficient, $f_{connected}$ is the interconnected fraction of the inundated areas, $W_{sfc}$ is the mass of surface water, $W_c$ is the mass of surface water when the storage is full, and $\Delta t$ is the model time step. $W_{sfc}$ is formulated as:

$$W_{sfc} = \frac{d}{2}\left(1 + \mathbf{erf}\left(\frac{d}{\sigma_{micro}\sqrt{2}}\right)\right) + \frac{\sigma_{micro}}{\sqrt{2\pi}}e^{\frac{-d^2}{2\sigma_{micro}^2}} \qquad \text{Eq. (5)}$$

where **erf** represents the error function, $d$ is the height of the surface water relative to the cell averaged elevation, and $\sigma_{micro}$ is the standard deviation of the microtopographic distribution that characterizes sub-grid elevation variation. Given the surface water height from the previous equation, the surface water fraction ($f_{h2osfc}$) of a cell is estimated with:

$$f_{h2osfc} = \frac{1}{2}\left(1 + \mathbf{erf}\left(\frac{d}{\sigma_{micro}\sqrt{2}}\right)\right)$$ 

Eq. (6)

The inundation areas are assumed to be randomly distributed within the grid cell, and the interconnected fraction of the inundated areas can be estimated based on percolation theory:

$$f_{connected} = \begin{cases} (f_{h2osfc} - f_c)^{\mu} & if\ f_{h2osfc} > f_c \\ 0, & if\ f_{h2osfc} \leq f_c \end{cases}$$

Eq. (7)

where $f_c$ is the threshold below which the inundated areas are not connected, and $\mu$ is a scaling exponent. The default parameter values in ELM of $f_c$ and $\mu$ are 0.4 and 0.14 for all the global cells, respectively.

The subsurface runoff is parameterized as an exponential function of water table depth and includes an ice impedance

factor ($\Theta_{ice}$) to account for the reduction in the bottom drainage when ice is present in the soil (Swenson et al., 2012):

$$R_{drai} = \Theta_{ice} q_{drai,max} \exp(-f_{drai} z_{\nabla})$$

Eq. (8)

$$\Theta_{ice} = 10^{-\Omega \frac{\theta_{ice}}{\theta_{sat}}}$$

Eq. (9)

where $q_{drai,max}$ is the maximum drainage rate, $f_{drai}$ is the decay factor, $\frac{\theta_{ice}}{\theta_{sat}}$ represents the ice-filled fraction of the pore space for the soil under the water table, and $\Omega$ is an adjustable parameter.

We follow the work of Huang et al. (2013) in selecting uncertain parameters and their corresponding ranges (Table 1). Three additional parameters are included in this study for surface water storage drainage and impacts of ice to subsurface

runoff and soil water dynamics, which represent new features in ELM compared to CLM4.0 used in Huang et al. (2013). All the parameter prior distributions are assumed to be a uniform distribution.

**Table 1.** Uncertain parameters' information.

| Parameter | Definition | Default value | Priors |
|---|---|---|---|
| $f_{max}$ | Maximum saturated fraction for a grid cell [-] | Derived from high-resolution DEM | $U(0.01, 0.907)$ |
| $f_{over}$ | Decay factor for surface runoff [$m^{-1}$] | 0.5 | $U(0.1, 5)$ |
| $f_{drai}$ | Decay factor for subsurface runoff [$m^{-1}$] | 2.5 | $U(0.1, 5)$ |
| $q_{drai,max}$ | Maximum subsurface drainage rate [$kg \cdot m^{-2} \cdot s^{-1}$] | $5.5 \times 10^{-3}$ | $U(1 \times 10^{-6}, 1 \times 10^{-1})$ |

| | | | |
|---|---|---|---|
| $b$ | Clapp and Hornberger exponent [-] | Determined by plugging the soil type into the equations of means from Table 5 of Cosby et al. (1984). | Uniform distributions with $\pm 50\%$ of the means as the lower and upper bounds. |
| $\psi_s$ | Saturated soil matrix potential [mm] | | |
| $K_s$ | Hydraulic conductivity [$mm \cdot s^{-1}$] | | |
| $\theta_s$ | Porosity [-] | | |
| $f_c$ | Surface water fraction threshold for outflow [-] | 0.4 | $U(0.1, 0.7)$ |
| $\mu$ | Scaling exponent for estimating connected surface water fraction [-] | 0.14 | $U(0.04, 0.24)$ |
| $\Omega$ | Adjustable parameter for ice impedance factor [-] | 6 | $U(0.6, 60)$ |

## 2.2 Model configuration

We ran ELM globally at a spatial resolution of 0.5°×0.5° driven by the Global Soil Wetness Project forcing data set (GSWP3v1) from 1991 to 2010, featuring 3-hourly, 0.5°×0.5° global atmosphere forcing. GSWP3v1 has been dynamically downscaled and bias-corrected based on the reanalysis data of Compo et al. (2011). The default configuration of ELM was used with a 30 min time step. With the default configuration, the hydrologic representations of ELM are the same as those in CLM4.5, as new model features such as the variably saturated flow model and subgrid topography are not included. Except the uncertain parameters listed in Table 1, the default values of all other ELM parameters were used in this study.

## 3 Uncertainty quantification framework

A detailed derivation of the PCE-based uncertainty quantification framework and BCS method used in this work is presented in Sargsyan et al. (2014); Debusschere et al. (2016). In this study, we used the Uncertainty Quantification Toolkit (UQTk; Debusschere et al., 2004; Debusschere et al., 2016) that includes implementations of PCE construction with BCS and subsequent global sensitivity analysis. Only a brief description of constructing the PCE-based surrogate for the ELM simulations is summarized below.

## 3.1 Polynomial Chaos Expansion

Let $\mathcal{M}$ denote a physical model (e.g., ELM) with uncertain parameters $\boldsymbol{X}$, where $\boldsymbol{X} = [X_1, X_2, ..., X_D]$ and $D$ represents the total number of uncertain parameters. In this study, the uncertain parameters $\boldsymbol{X}$ are listed in Table 1 and $D$ is 11. A scalar Quantity of Interest (QoI), $\hat{y}$ (e.g. runoff at a specified time from a specified location), obtained using a sample of random parameters, $\boldsymbol{x}$, can be expressed as a polynomial expansion:

$$\hat{y} = \mathcal{M}(\boldsymbol{x}) = \sum_{\alpha} c_\alpha \Psi_\alpha(x) \qquad \text{Eq. (10)}$$

where $\Psi_\alpha$ is a polynomial and $c_\alpha$ is the corresponding coefficient. In practice, $\boldsymbol{x}$ is scaled to [-1 1] from the original uncertainty input range. The polynomial expansion in Eq. (10) is written with respect to multivariate orthogonal polynomials:

$$\Psi_\alpha(x) = \prod_{i=1}^{D} \Psi_{\alpha_i}(x_i) \qquad \text{Eq. (11)}$$

where $\Psi_{\alpha_i}(x_i)$ is a univariate polynomial, whose form is associated with the prior distribution of uncertain input variable $X_i$ (e.g., Legendre polynomials are used when the input variable follows a uniform distribution), and $\alpha_i$ is a member of the multi-index $\boldsymbol{\alpha} = [\alpha_1, \alpha_2, ..., \alpha_D]$, which represents the degrees of the univariate polynomial terms. Readers should refer to Dwelle et al. (2019) for details about the selection of polynomial terms and an illustration of how the multi-index is used to construct a PCE-based surrogate. In practice, Eq (11) is approximated with a truncated PCE by only selecting terms with a total degree of polynomials smaller than a certain value $p$ (Xiu and Karniadakis, 2002; Lin and Karniadakis, 2009). This leads to a finite set $\mathcal{A}_p = (\boldsymbol{\alpha}: \sum_{i=1}^{D} \alpha_i \le p)$ for the multi-index $\boldsymbol{\alpha}$ to take:

$$\mathcal{M}(\boldsymbol{x}) \approx \mathcal{M}^{PC}(\boldsymbol{x}) = \sum_{\alpha \in \mathcal{A}_p} c_\alpha \Psi_\alpha(x) = \sum_{j=0}^{P} c_j \Psi_j(x) \qquad \text{Eq. (12)}$$

where j represents the counter index of any possible multi-index $\boldsymbol{\alpha}$ in $\mathcal{A}_p$ in a predefined order (see details in Appendix B of Dwelle et al. (2019)). The coefficients ($c_j$) for the $P + 1$ polynomial bases are computed using training simulations of $\mathcal{M}(\boldsymbol{x})$ (e.g., ELM) to construct the truncated PCE approximation in Eq (12). The number of the polynomial basis is determined by both the input dimension $D$ and the total degree for truncation $p$ (Xiu and Karniadakis, 2002):

$$P + 1 = \frac{(D+p)!}{D!\,p!}, \qquad \text{Eq. (13)}$$

The value $P$ increases rapidly as the number of uncertainty input variables increases. For example, 11 uncertain parameters (e.g., $D = 11$) with a truncated PCE order of $p = 4$ leads to **1,365** coefficients to solve in Eq (12). It is computationally prohibitive to run 1,365 global ELM simulations, so we adopted the BCS method of Sargsyan et al. (2014) that requires a much smaller number of ELM simulations to construct a PCE-based surrogate. The BCS method computes only a sparse set of $c_j$ to construct the surrogate of a form given by Eq (12) because not all $\Psi_j(x)$ are relevant for the given QoI (Sargsyan et al., 2014).

## 3.2 Global sensitivity analysis

In this study, we performed variance-based, global sensitivity analysis using Sobol indices (Sobol′, 2001). For PCE-based surrogate model, the main Sobol index, $S_i$, for the uncertain parameter $X_i$ can be estimated as:

$$S_i = \frac{\sum_{j \in \Pi_i} c_j^2 \, ||\Psi_j||^2}{\sum_{j=0}^{P} c_j^2 \, ||\Psi_j||^2},\tag{Eq. (14)}$$

where $\Pi_i$ denotes all the indices of polynomial basis terms in Eq (12) that only involve parameter $X_i$, and $||\Psi_j||$ is the norm of the polynomial $\Psi_j(x)$. The main Sobol index $S_i$ can be interpreted as the fraction of variance in the output that is associated with the uncertainty model parameter $X_i$ only when other parameters are fixed at constant values. Similarly, one can estimate the Sobol index for any pair of parameters $X_i$ and $X_{i'}$ to represent parameter interaction sensitivity with the coefficients $c_j$ (Sargsyan et al., 2014).

## 3.3 Parameter inference

Parameter inference is used to determine a set of model parameters that reduces the error between observation and model prediction. The model inverse problem can be solved with the Bayes theorem:

$$p(\boldsymbol{X}|\boldsymbol{y}) = \frac{L(\boldsymbol{y}|\boldsymbol{X})p(\boldsymbol{X})}{p(\boldsymbol{y})},\tag{Eq. (15)}$$

where $p(\boldsymbol{X}|\boldsymbol{y})$ is the posterior distribution of parameter $\boldsymbol{X}$ given observation $\boldsymbol{y}$, $L(\boldsymbol{y}|\boldsymbol{X})$ is the likelihood function, $p(\boldsymbol{X})$ represents the prior distribution of $\boldsymbol{X}$, and $p(\boldsymbol{y})$ is merely a normalizing constant for the purposes of parameter calibration. The discrepancy between the model and observations, $\boldsymbol{\epsilon} = \boldsymbol{y} - \mathcal{M}(\boldsymbol{X})$, should be included in the likelihood function. It is common to assume the error term (e.g., $\boldsymbol{\epsilon}$) follows a Gaussian distribution with vanishing mean:

$$\epsilon_i \sim \mathcal{N}(0, \sigma^2), i = 1, 2, \dots, N\tag{Eq. (16)}$$

where $N$ is the number of observations used to infer the parameters (e.g., time series of monthly runoff), and the standard deviation, $\sigma$, can be inferred from the data (see Sec 3.4). Then, the likelihood function can be written as:

$$L(\boldsymbol{y}|\boldsymbol{X}) = \prod_{i=1}^{N} \frac{1}{\sqrt{2\pi\sigma^2}} \exp\left[-\frac{\left(y_i - \mathcal{M}_i(\boldsymbol{X})\right)^2}{2\sigma^2}\right],\tag{Eq. (17)}$$

The logarithm of Eq (17) leads to the least-squares objective function that is used for deterministic parameter estimation in practice (Sargsyan et al., 2015):

$$\log L(\boldsymbol{y}|\boldsymbol{X}) = -\sum_{i=1}^{N} \frac{\left(y_i - \mathcal{M}_i(\boldsymbol{X})\right)^2}{2\sigma^2} - \frac{N}{2}\log(2\pi\sigma^2),\tag{Eq. (18)}$$

The posterior distribution in Eq (18) is difficult to compute in practice, hence we estimate it through samples obtained by the Markov Chain Monte Carlo (MCMC) method. Specifically, 1,000 iterations are used as the "burn-in" period in this study and the sampling of the posterior distribution is saved every 10 iterations. We run MCMC for 10,000 steps, resulting in 1,800

samples to construct the posterior distribution. We have employed adaptive MCMC method of Haario et al. (2001), in which
the parameter space is searched according to proposal steps with a covariance that is updated on-the-fly.

## 3.4 Quantity of interest

In this study, the physical model $\mathcal{M}$ and the QoI $\hat{y}$ correspond to ELM and runoff, respectively. The development of
a surrogate model for the simulated runoff for each grid cell for each month of a 20-year simulation would require 240 (= 12
months × 20 years) PCE-based surrogates. Although developing a PCE-based surrogate is not expensive, it is computationally
expensive to train 240 PCEs for each of the 70302 grid cells in the global domain. The parameter inference process for 240
PCEs for each grid cell will further increase the computational cost. We reduce the number of QoIs by training the surrogate
model for the root mean square error (RMSE) between simulated runoff and observations instead of training the surrogate
model to predict monthly runoff. The RMSE is given as:

$$RMSE = \sqrt{\frac{1}{N}\sum_{i=1}^{N}\left(R_i^{sim} - R_i^{obs}\right)^2}, \qquad\qquad \text{Eq. (19)}$$

where $R_i^{sim}$ and $R_i^{obs}$ represent grid-level simulated total runoff and observed total runoff, respectively, for $i$-th month in the
simulated period, and $N$ represents the number of simulation months. Consequently, only one surrogate model is needed for
each grid cell to quantify the performance of ELM in capturing the monthly runoff variation for a given uncertain parameter
set. The selection of RMSE as QoI in constructing surrogate models significantly reduce the computational burden of
surrogates' construction and parameter inference. We performed ELM simulations using 200 parameter sets that were
randomly sampled from the range specified in Table 1. A set of 175 ELM simulations were used for training the surrogate
models and the other 25 simulations were used for validating their performances. The performance of the PCE-based surrogate
model can be affected by the truncated order (Dwelle et al., 2019). For each grid cell, we train the surrogate with $p = 1, 2, \dots, 7$
separately, and picked the order that minimizes the relative norm-2 error (RE) of validation simulations:

$$RE = \frac{||RMSE_{val}^{PC} - RMSE_{val}^{\mathcal{M}}||_2}{||RMSE_{val}^{\mathcal{M}}||_2}, \qquad\qquad \text{Eq. (20)}$$

where $RMSE_{val}^{PC}$ and $RMSE_{val}^{\mathcal{M}}$ represent the PCE-simulated and ELM-simulated vector of $RMSE$ of the 25 validation
simulations, respectively. Then, the trained surrogate models, $RMSE^{PC}$, can be plugged into the likelihood function of Eq (18)
seamlessly:

$$\log L(\boldsymbol{y}|\boldsymbol{X}) = -\frac{N \cdot (0 - RMSE^{PC})^2}{2\sigma^2} - \frac{N}{2}\log(2\pi\sigma^2), \qquad\qquad \text{Eq. (21)}$$

The standard deviation of error between model simulated runoff and observation exhibits significant monthly variation. To
provide a reasonable value of $\sigma$, we further assume $\sigma$ in Eq (21) has a different meaning than that in Eq (18) by taking RMSE
as model simulation, and 0 to as the target. Therefore, $\sigma$ is approximated as the standard deviation of the difference between 0
and RMSEs, where each RMSE was calculated using simulated runoff and observation for a given training simulation. Our

estimation of $\sigma$ leads to a reasonable posterior (see Sec 5.4), though other methods can also be used to estimate $\sigma$. We acknowledge that the value of $\sigma$ may have an impact on the parameter posteriors, but investigating the sensitivity of $\sigma$ on the posteriors is beyond the scope of this study.

### 3.5 Calibration procedure

In summary, the following procedures were implemented to determine the optimal parameter values and their joint probability distribution:

1. Run ELM with 200 parameter sets randomly sampled with the range specified in Table 1. Construct PCE-based surrogate models to mimic the RMSE between the ELM and GRUN runoff dataset with 175 simulations and validate the performance of the surrogate models with the other 25 simulations.

2. Implement sensitivity analysis with the surrogate models to reduce parameter dimensionality for calibration by ignoring the parameters with negligible Sobol index (e.g., less than 0.05).

3. Estimate the Bayesian posterior of the most sensitive parameters for each grid through MCMC process with the runoff dataset of Ghiggi et al. (2019).

4. It has shown small surrogate error can result in significant deviation of the inferred parameter (Laloy and Jacques, 2019). To further search the optimal parameters and construct the runoff posterior uncertainty, we ran ELM simulations with additional 100 samples from the posteriors of the 3 most sensitive parameters for all global grid cells and default values were used for less sensitive parameters.

5. The parameters with the minimum RMSE between simulations and reference runoff data from the 100 ELM simulations were determined as the optimal parameter value for each grid cell.

### 4 Data

#### 4.1 Observation-based runoff data

The $0.5° \times 0.5°$ global observed-based runoff (GRUN) dataset of Ghiggi et al. (2019) was used in this study as the observation within the calibration framework for parameter inference. The GRUN dataset was generated from a trained random forests (RF) model (Breiman, 2001) that used precipitation and near-surface temperature to predict monthly runoff. The training runoff data were derived from Global Streamflow Indices and Metadata Archive (GSIM; Gudmundsson et al., 2018; Do et al., 2018), and only the gauges with contributing area comparable to cell area of $0.5° \times 0.5°$ were used. GSWP3 atmospheric forcing was used for training and reconstruction of the monthly global runoff.

#### 4.2 Model benchmarks

The ILAMB package (Collier et al., 2018) was used to evaluate the simulated water and energy cycles from the calibrated ELM against various benchmarks. Specifically, a gridded energy flux data (FLUXCOM; Jung et al., 2019) that was

generated by machine learning with flux tower measurements was used to evaluate latent and sensible heat fluxes; Global Land Evaporation Amsterdam Model version 3 (GLEAMv3; Martens et al., 2017) product was used to evaluate global ET; Gravity Recovery And Climate Experiment (GRACE; Kim et al., 2009) data were used to evaluate terrestrial water storage anomaly (TWSA). Details about ILAMB can be found at https://www.ilamb.org.

The inter-Sectoral Impact Model Intercomparison Project (ISIMIP) archived simulations from multiple global hydrological models and land surface model forced by the same atmosphere forcings (Warszawski et al., 2014). We used 13 available models from the second phase water sector (ISIMIP2a; Gosling et al., 2019) to provide a benchmark for the uncertainty of annual runoff magnitude and trend. Only the models in ISIMIPI2a that were driven by the GSWP3 forcing without accounting for human activity impacts were selected here to be consistent with ELM's configuration.

## 4.3 Evaluation metrics

Two metrics were used to evaluate ELM's performance of simulating runoff at monthly scale with calibrated parameters, including the Nash-Sutcliffe efficiency (NSE; Nash and Sutcliffe, 1970) and the Kling-Gupta Efficiency (KGE; Gupta et al., 2009) which are computed as

$$NSE = 1 - \frac{\sum_{i=1}^{N}\left(R_i^{sim} - R_i^{obs}\right)^2}{\sum_{i=1}^{N}(R_i^{obs} - \mu_{obs})^2}, \qquad \text{Eq. (22)}$$

$$KGE = 1 - \sqrt{(\rho - 1)^2 + \left(\frac{\sigma_{sim}}{\sigma_{obs}} - 1\right)^2 + \left(\frac{\mu_{sim}}{\mu_{obs}} - 1\right)^2}, \qquad \text{Eq. (23)}$$

where $R_i^{sim}$ and $R_i^{obs}$ represent cell-level simulated total runoff and observed total runoff, respectively, for the $i$-th month, $\mu_{obs}$ is the corresponding averaged observation, $\rho$ is the correlation coefficient between simulation and observation, $\sigma_{sim}$ is the standard deviation in simulations, $\sigma_{obs}$ is the standard deviation in observations, and $\mu_{sim}$ is the simulation mean. Both NSE and KGE vary from $-\infty$ to 1, and a perfect model performance is indicated by NSE = 1 and KGE = 1. NSE < 0 and KGE < -0.41 mean the simulations are worse estimates than the mean of observations, indicating a bad model performance (Knoben et al., 2019).

The sensitivity of runoff to precipitation is a critical aspect for runoff simulation evaluation, considering changes in precipitation will continue in the future (Trenberth, 2011). Therefore, we evaluated the sensitivity of runoff to the precipitation anomalies with the calibrated parameters. The sensitivity was quantified by the slope of linear regression ($\beta$) between runoff anomalies ($\Delta R$) and precipitation anomalies ($\Delta P$):

$$\Delta R = \beta \Delta P + \epsilon, \qquad \text{Eq. (24)}$$

The interception $\epsilon \approx 0$, implies the mean runoff is related to mean precipitation.

We also evaluated the impacts of parameters on the runoff trend. Specifically, the magnitude of runoff trend was calculated with Sen's slope (Sen, 1968), which is nonparametric and not sensitive to the outliers. Then, Mann-Kendall test was used to determine if the trend is significant or not at confidence level $\alpha = 0.05$.

# 5 Results

## 5.1 Refinement of $f_{drai}$ for arid region

The proposed prior for $f_{drai}$ is not suitable for all the climate regions, such as simulations with the full the range of
$f_{drai}$ defined in Table 1 results in unrealistic runoff for arid regions. For example, the simulated runoff from an example grid cell with $f_{drai} < 0.4$ shows higher magnitudes and lower variabilities compared to simulations with $f_{drai} \geq 0.4$ (Figure 1a). Lower $f_{drai}$ can lead to unrealistically high subsurface runoff according to the exponential function of baseflow drainage (Eq (8)) for the arid regions, where the precipitation is not enough to maintain the water table at a reasonable level. Such simulations with $f_{drai} < 0.4$ result in high nonlinearity in the simulated runoff, and hence the PCE-based surrogate model cannot capture
the model behaviours (Figure 1b). The performance of surrogate models is improved by constraining the lower bound of $f_{drai}$ to 0.4 (Figure 1c). Therefore, $f_{drai}$ is refined as [0.4 5] for areas that are identified as arid climate in the Köppen climate classification (Figure S1), and [0.1 5] is used in other regions.

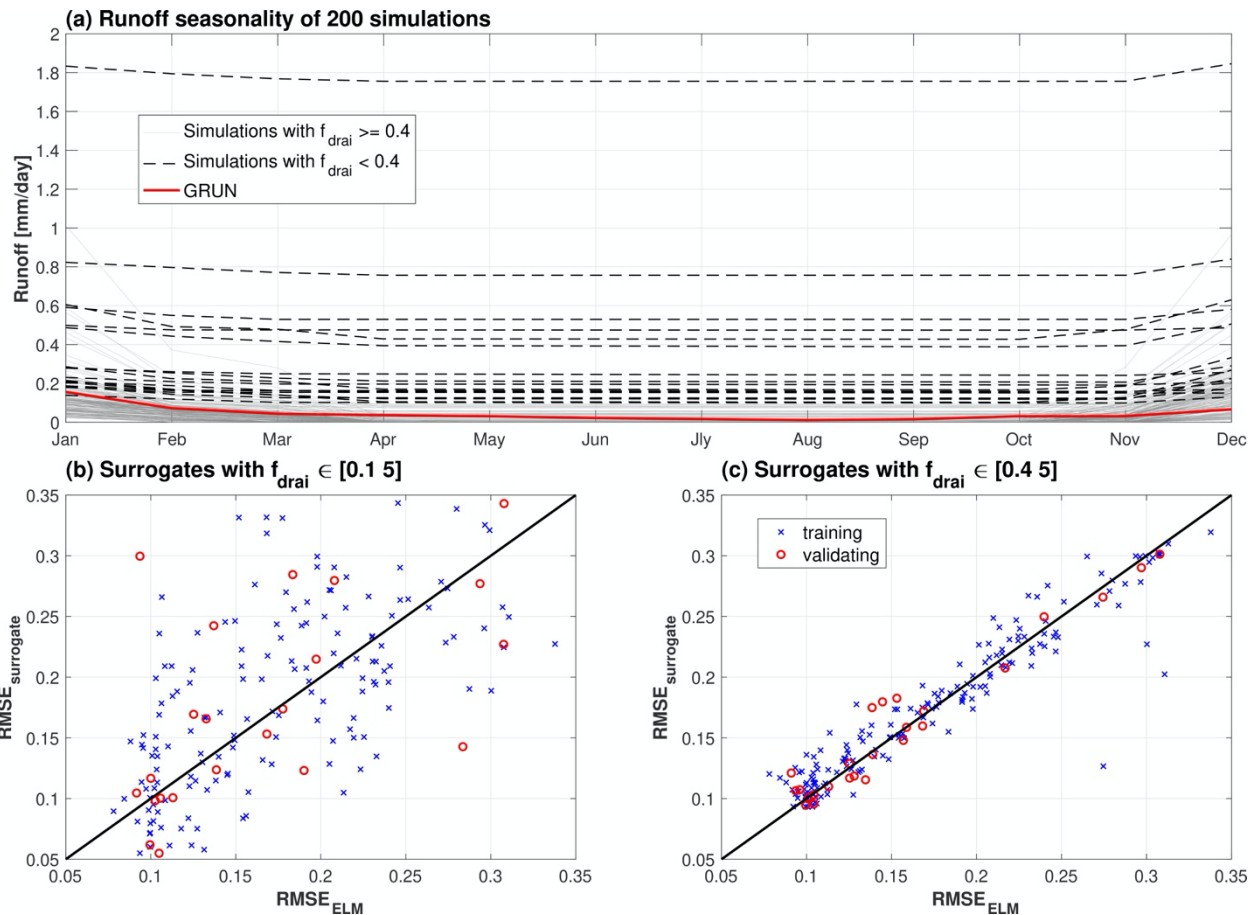

**Figure 1.** Validation of surrogate performance for an example grid cell from arid region. Subplot (a) shows runoff seasonality from all the 200 simulations with samples from parameter priors. Subplot (b) shows the validation of the surrogate model trained with original ranges of $f_{drai}$ given in Table 1 in main text. Subplot (c) shows the validation of the surrogate model trained with constrained $f_{drai}$.

## 5.2 Validation of surrogate models

The PCE-based surrogate models can mimic the variations of RMSE between ELM-simulated runoff and the GRUN runoff with the truncated order determined in Figure S2. Specifically, the surrogate models exhibit good performance for the validation simulations with $RE < 0.1$ for 70% of the global domain (Figure 2a). The global averaged RE of surrogate models for the validation simulations is around 0.1, with the largest error over the arid regions (Figure 2b). While 41% of the arid region shows an acceptable performance in the surrogate models when narrowing the range of $f_{drai}$ with RE less than 0.15, the RE of other arid areas remain high (Figure 2a). Additional simulations were performed to investigate if the lower performance of surrogate models for arid regions is due to insufficient number of training simulations. We randomly selected 20 grid cells from the arid region and ran 2,000 ELM simulations with random samples from the parameter priors as summarized in Table 1. The RE of surrogate models for the 20 grid cells remained large (e.g., Re > 0.2) even as the number of training simulations were increased (Figure S3). Thus, the lower performance of surrogate models over the arid regions is not dependent on the number of training simulations.

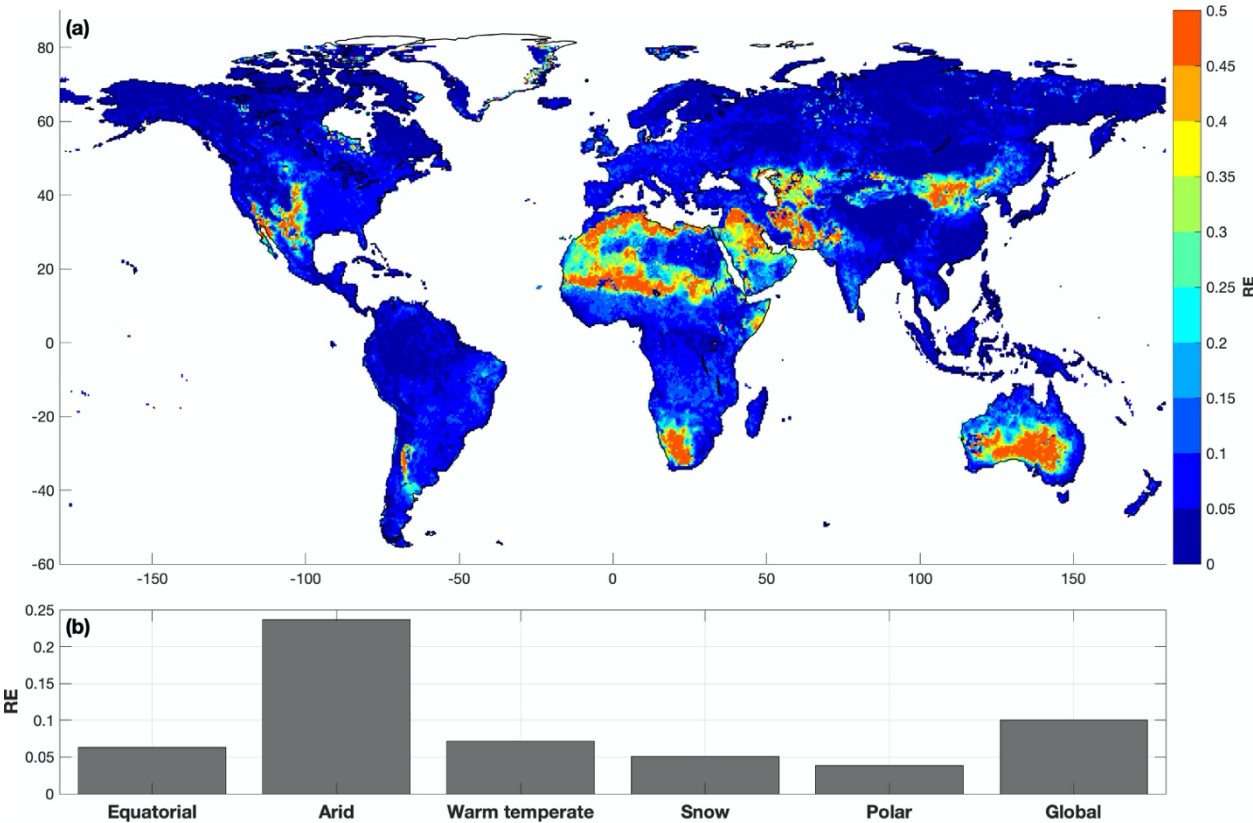

**Figure 2.** Relative norm-2 error of the surrogate models for the validation simulations. Subplot (a) shows the spatial distribution of the errors, and subplot (b) shows the average errors for the grid cells in each climate defined by Köppen climate classification.

Most surrogate models with large RE are in extremely dry arid region; for example, RE > 0.2 are mainly from grid cells with annual runoff < 0.05 mm/day (Figure 3). The RE of surrogate models tends to decrease for areas with relatively higher annual runoff that are still from arid region (annual runoff < 0.5 mm/day in Figure 3). However, the runoff uncertainty in extremely dry areas will have negligible impact on the global water cycle. Surrogate model with RE > 0.15 is considered as not sufficiently accurate and such grid cells are excluded in the sensitivity analysis presented next.

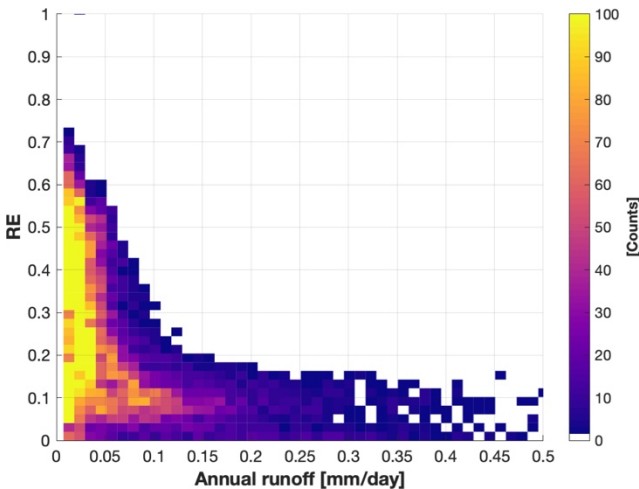

**Figure 3.** Plot of relative norm-2 error (RE) of surrogate models for the validation simulations vs. averaged annual runoff magnitude with all the grid cells from arid region.

## 5.3 Global sensitivity analysis

The most significant ELM parameters identified for runoff generation are $f_{over}, f_{drai}, \psi_s, f_c$, and $\Omega$ based on the spatial distribution of the main Sobol indices (Figure 4), while the other 6 parameters have negligible contributions to the runoff variations (Figure S4). In equatorial regions, $f_{drai}$ and $f_{over}$ are equally sensitive and account for 39% and 36% of the average runoff variations, respectively, as indicated by the size of circles in Figure 5a; while $\psi_s$ is the secondary sensitive parameter. For the arid regions, $f_{drai}$ is the most sensitive parameter, and $f_{over}, f_c, K_s$, and $\psi_s$ are secondary sensitive parameters with a similar value for the main Sobol indices (Figure 5b). Although other parameters show negligible main Sobol indices for arid regions, they have shown sensitivities when interacting with each other as denoted by the thickness of the lines between each pair of parameters in Figure 5b. The complex joint sensitivity results in high nonlinearity in the runoff variations, representing a possible reason for the poor performance of PCE for arid regions. The most significant uncertain parameters for the warm temperate region are the same as those for the equatorial region (Figure 5c). Snow and polar climates have similar sensitivity pattern, with $f_c$ and $\Omega$ are the two most important uncertain parameters (Figure 5d and e). In colder region, the contribution of surface water storage drainage, which is controlled by $f_c$, is large to the total runoff because of prominent surface water areas (Pekel et al., 2016). The hydraulic conductivity and groundwater drainage when ice is present in the soil is controlled by $\Omega$, which has a significant impact on runoff generation process when the soil is partial or fully frozen. The surface water storage and ice impedance factor, which were not included in the version of the model used in previous study (Hou et al., 2012; Huang et al., 2013), are found to be the most sensitive parameters in cold regions. Besides arid region, other regions show smaller sensitivities to parameter interactions.

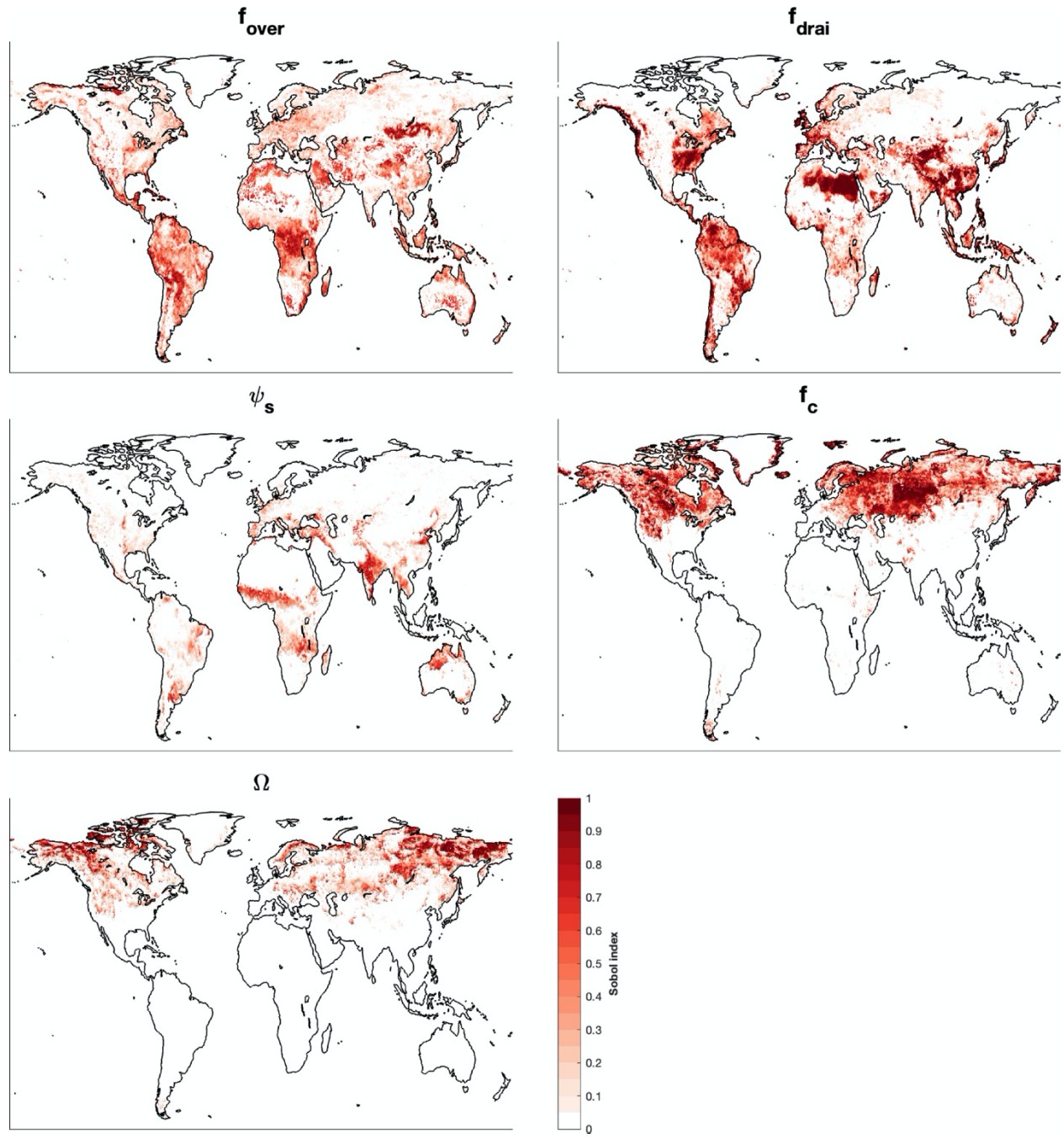

**Figure 4.** Spatial distribution of main Sobol index for the sensitive parameters.

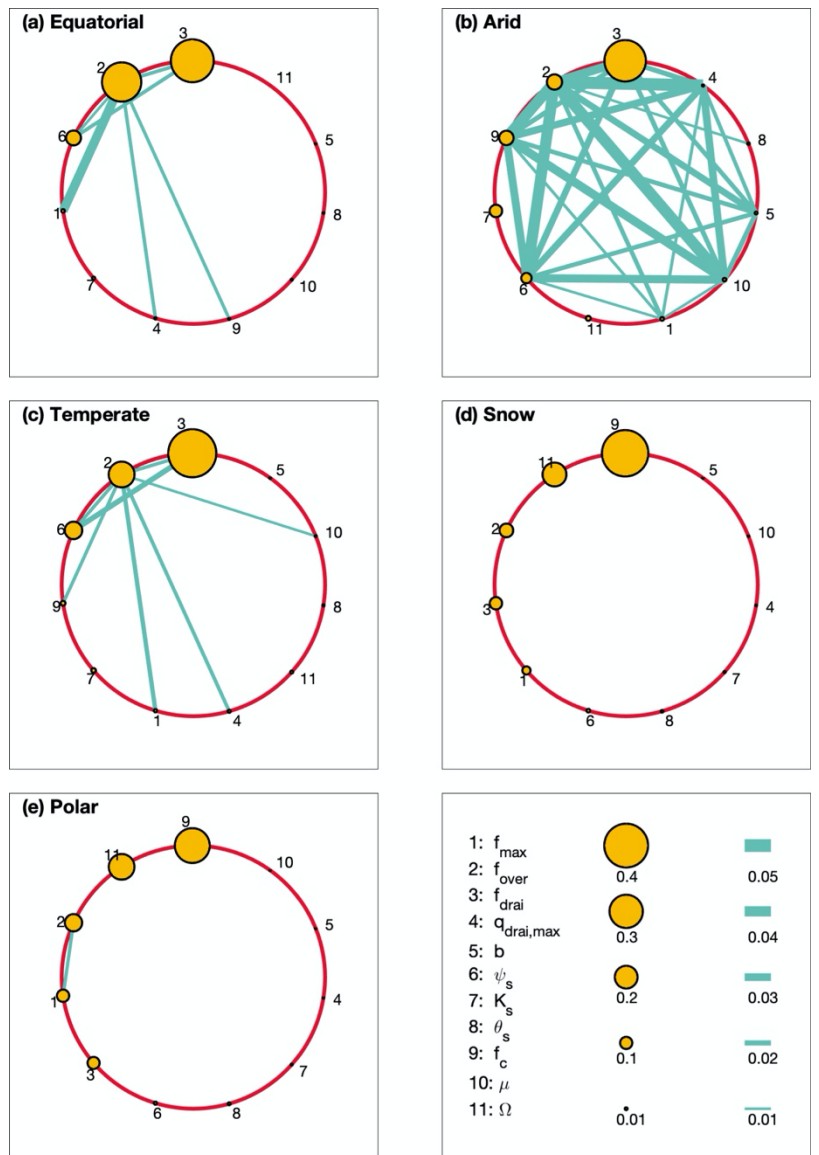

**Figure 5.** Averaged main Sobol index and joint Sobol index for different climates defined by Köppen climate classification. Only the cells with the relative norm-2 errors of PCE-based surrogate models for validating simulations less than 0.15 are used in estimating the averaged sensitivity for each climate region. The size of the circles and thickness of the lines are proportional to main Sobol index and joint Sobol index, respectively. The legend in the right bottom subplot shows the Sobol index for the

355 corresponding size of circle and thickness of line.

## 5.4 Parameter dimensionality reduction

The ELM simulated runoff is significantly sensitive to three or fewer parameters with Sobol index $> 0.05$ for 81.3% of the total grid cells (Figure S5). Therefore, we sampled only the three most sensitive parameters in each grid cell in the MCMC process to perform parameter inference as mentioned in Sec 3.3. The posteriors of the three calibrated parameters $(f_{drai}, f_{over}, \psi_s)$ at an example grid cell $(56.75°W, 11.25°S)$ are much more constrained than the priors after the MCMC simulation with the surrogate model (Figure 6a, b, and c). The third parameter, $\psi_s$, has a relatively wider posterior than the first two parameters because its sensitivity is much smaller (e.g., Sobol index $= 0.08$). The Gelman-Rubin R statistic of Gelman and Rubin (1992) computed with 5 MCMC chains (after burn-in period) is 1.002, 1.004, 1.003 for $f_{drai}, f_{over}$, and $\psi_s$, respectively, suggesting our MCMC simulation has converged (see convergence curve in Figure S6). ELM simulations with a large number of samples from parameter priors are needed to identify the optimal parameter that minimizes RMSE, for example, 10,000 surrogate simulations are used to find the parameters that yield $RMSE = 1$ (Figure 6d). In contrast, due to the reduced parameter dimensionality and narrowed range, much fewer samples (e.g., 100) are needed to find the better parameter values (e.g., corresponding to $RMSE < 1$) when they are sampled from the parameter posteriors (Figure 6d). The spatial distribution of the parameter values at 5% and 95% of the posteriors is shown in Figure S7 and Figure S8, respectively.

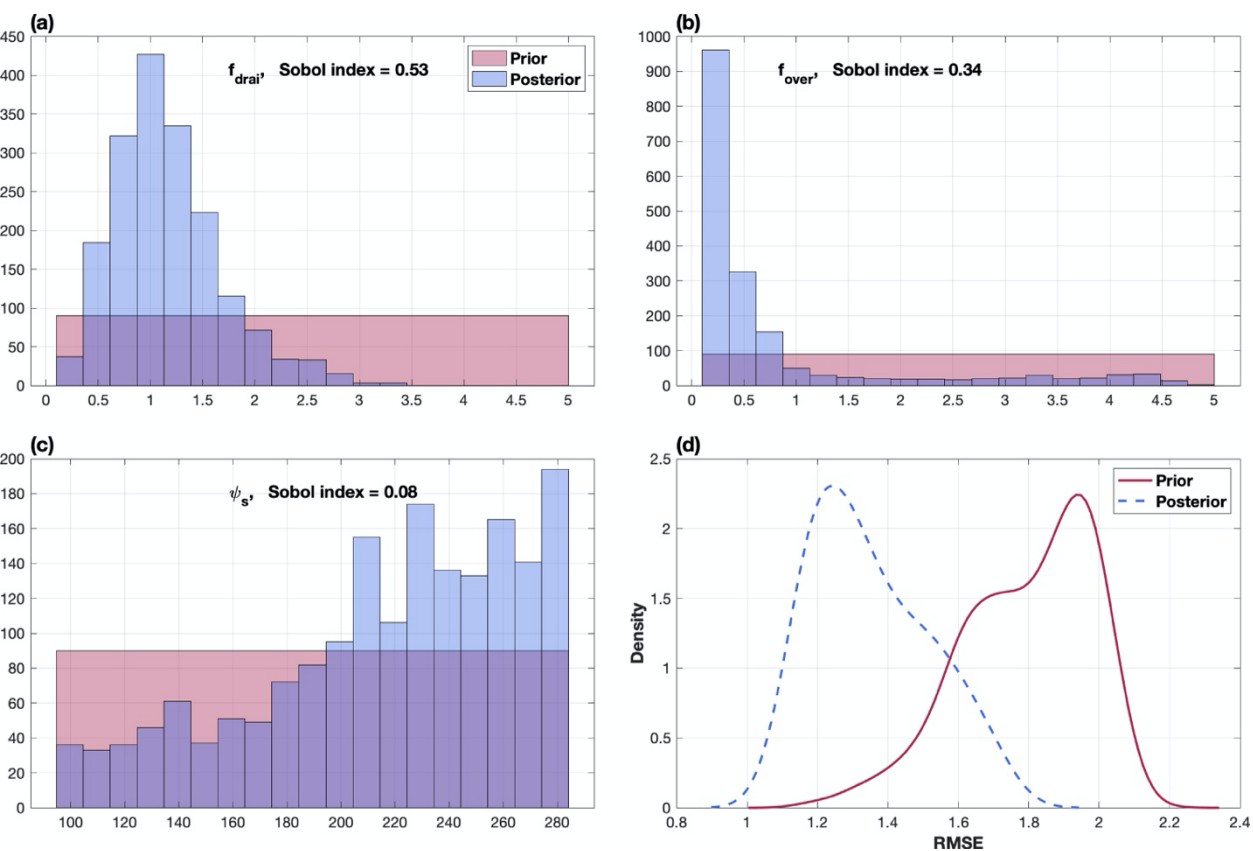

**Figure 6.** Posteriors of (a). $f_{drai}$, (b). $f_{over}$, and (c). $\psi_s$ from parameter inference process at an example grid cell. Subplot (d) shows the probability density function (PDF) of RMSE evaluated with surrogate models forced by 100 samples from parameter posteriors and 10,000 samples from parameter priors.

## 5.5 Optimal parameter values

The procedure described in Sec 3.5 is used to find the optimal parameter values for the three most sensitive parameters for each grid cell. For the grid cells with RE > 0.15 for surrogate models, the optimal parameter value is determined from the training and validation simulations (e.g., 200 simulations with random parameter values from priors) that yield minimum RMSE. The optimal parameter values show clear regional patterns (Figure 7). Specifically, the optimal $f_{over}$ tends to be lower than the default value for the equatorial and partial snow areas (Figure 7a). The optimal $f_{over}$ is found to be higher than the default value for the arid areas, while it is around the default value on average for the warm temperate areas (Figure 7a). For the same water table depth, lower $f_{over}$ leads to higher saturation fraction (Eq. 3), that in turn leads to larger surface runoff (Eq. 2). The calibrated $f_{drai}$ is lower than the default value for both equatorial and arid regions (Figure 7b). The optimal $f_{drai}$ for warm temperate areas show different patterns, with higher values over eastern US and Europe, but lower values over South-eastern China. The generation of subsurface runoff depends on $f_{drai}$ (Eq. 8) with lower $f_{drai}$ leading to larger subsurface runoff. $\psi_s$ affects the runoff generation through its impact on soil water movement, such as the soil water flux is larger at saturation with higher $\psi_s$. As shown in Figure 7c, higher $\psi_s$ are needed to minimize the RMSE for all regions that show sensitivity to this parameter, except some grid cells from polar area. Over the high latitudes of Northern Hemisphere, higher $f_c$ and lower $\Omega$ are found in the optimal parameters (Figure 7d, e). The surface water storage can store more water at higher $f_c$ by reducing surface water runoff (Eq. 4, 7), thus leading to a lower and delay peak runoff than the default values. Further, the lower $\Omega$ values have less impacts of ice on hydraulic conductivity (Eq. 7.89 in Olson et al. (2016)) and drainage (Eq. 9), which leads to higher runoff for the winter seasons.

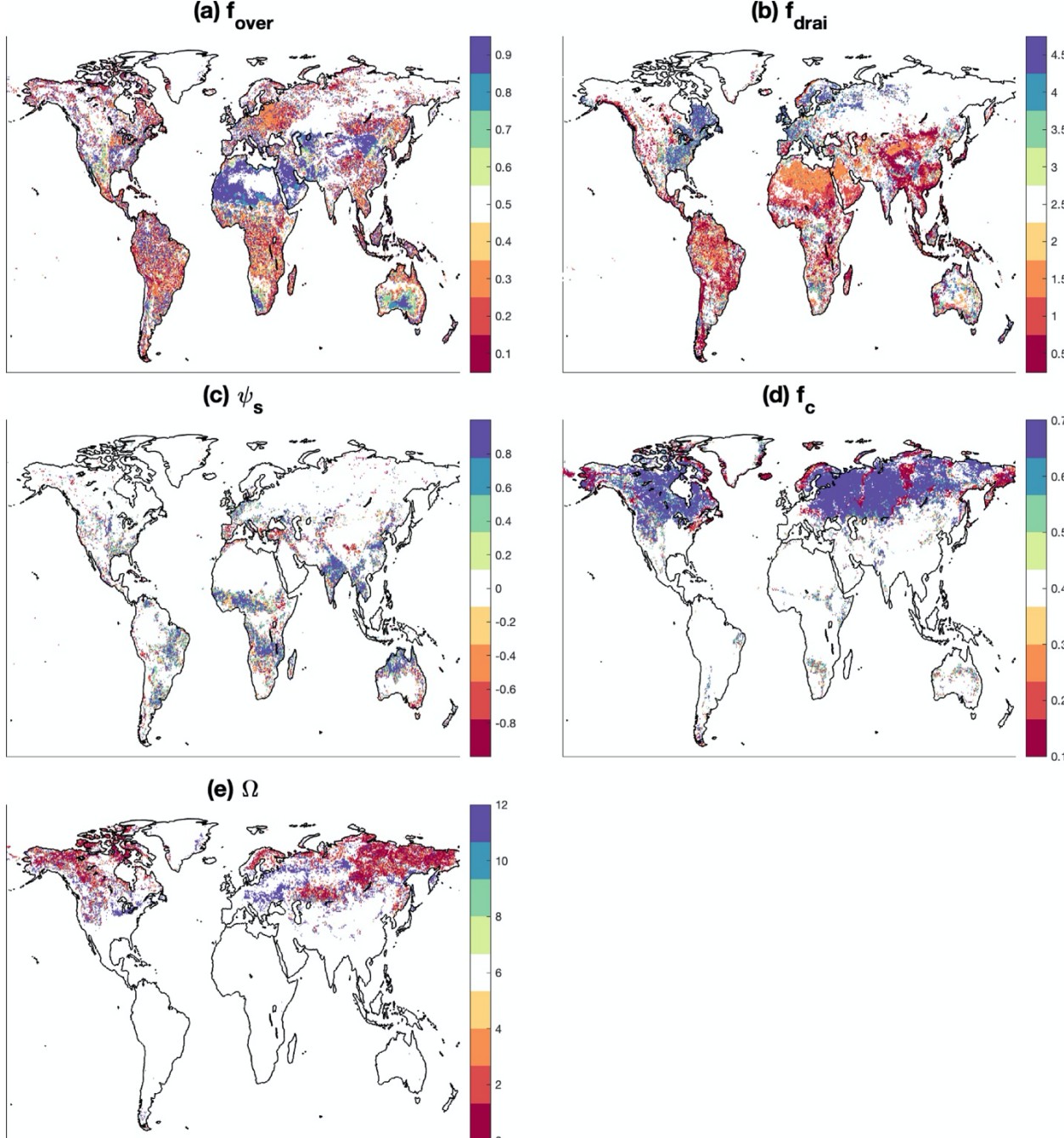

**Figure 7.** Optimal values for the sensitive parameters. The default values for the parameters are defined at the midpoint of the colormap. There are no certainty bounds for $\psi_s$ from different grid cells because it is determined by the soil properties. Therefore, the values of $\psi_s$ are scaled to [-1, 1] in subplot (c) for each grid cell with the corresponding upper bound ($\psi_{s,max}$) and lower bound ($\psi_{s,min}$): $\frac{2}{\psi_{s,max}-\psi_{s,min}}\psi_s - \frac{\psi_{s,max}+\psi_{s,min}}{\psi_{s,max}-\psi_{s,min}}$.

### 5.6 Evaluation of ELM with the optimal parameters

The ELM-simulated runoff with the optimal parameter values shows improved skills of capturing the spatiotemporal variation of monthly runoff at global scale with higher NSE and KGE compared to the simulation with default parameter values (Figure 8). Specifically, the median of NSE and KGE from all global grid cells increases from -0.88 and -0.05 to 0.06 and 0.31, respectively. Over the western US coast, southeast and Midwest of US, western Europe, equatorial areas, the performance of the calibrated ELM is better with NSE > 0.5 and KGE > 0.7. While the performance of other areas (e.g., western US, Sahara and Arabian desert, central and eastern Asia, and partial high latitude regions) is improved compared to simulations with the default parameter values, the NSE and KGE still have negative values. The higher model errors in those regions cannot be resolved by calibration as 1) the simulation resolution is too coarse to resolve the topographic impacts (Chegwidden et al., 2020); 2) the snow melting processes are not calibrated in this study, and the onset of snowmelt in ELM is poorly represented (Toure et al., 2018); and 3) hydrology of arid areas is not well understood (Pilgrim et al., 1988). Except for the calibration period, ELM with the optimal parameters also shows an improved performance in runoff simulation for another period (2011-2013) (Figure S9).

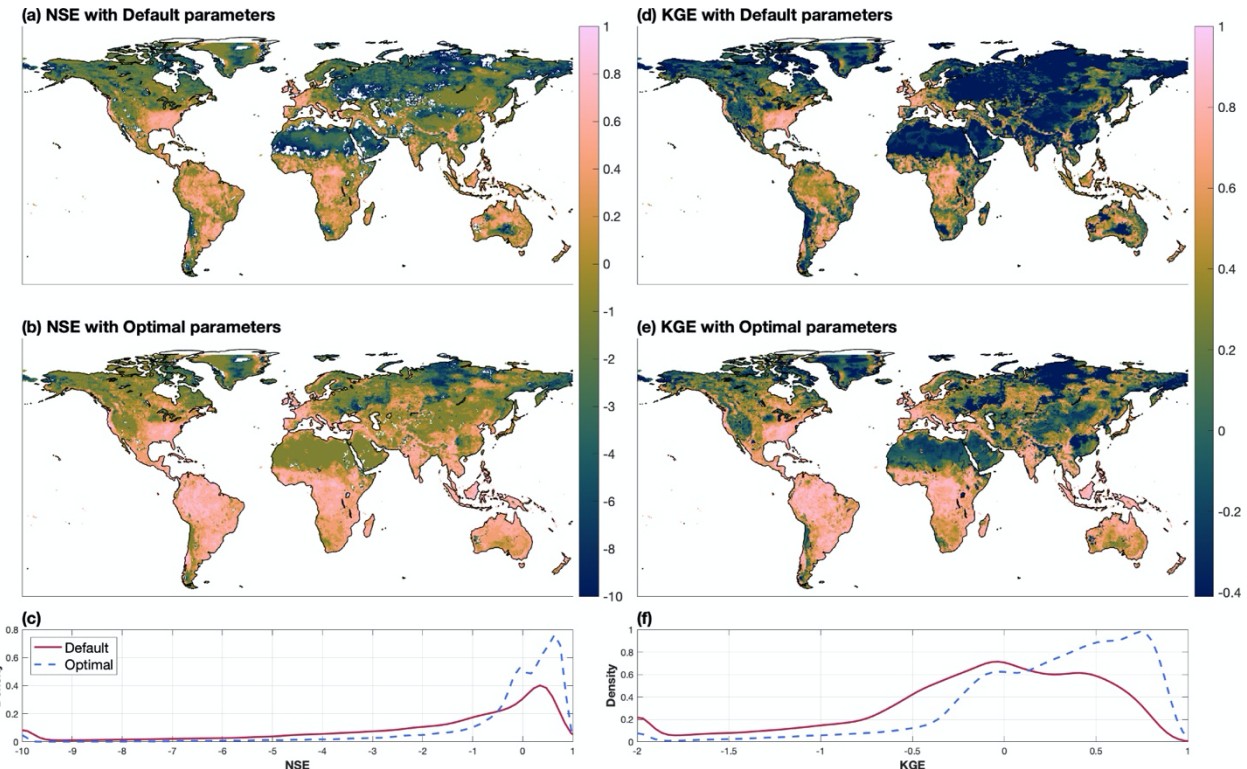

**Figure 8.** Evaluation of simulated monthly runoff for 1991-2010 at grid level with default and optimal parameters. Subplot (a) and (b) show the NSE metrics between the GRUN runoff and simulated runoff with default and optimal parameter, respectively. Subplot (c) shows the comparison of the probability density function (PDF) of NSE metrics from all the global grid cells. Subplot (d), (e), and (f) illustrate the evolution with KGE metric.

Compared to the reference runoff (Figure 9a), the ELM simulation with default parameter values tends to overestimate the sensitivity of runoff to precipitation ($\beta$ in Eq (24)) for the equatorial and arid regions, but underestimates $\beta$ in the warm temperate regions, such as eastern US, China, and eastern coasts of Australia (Figure 9b). The simulation with optimal parameter values is able to more accurately estimate $\beta$ than the simulation with default parameter values with improved spatial correlation coefficient from 0.22 to 0.56, and lower RMSE from 1.22 to 0.65 (Figure 9c). However, some significant discrepancy of $\beta$ still exists in the simulation with optimal parameter values (e.g., eastern China), implying the sensitivity is not well constrained in ELM for certain regions even after model calibration.

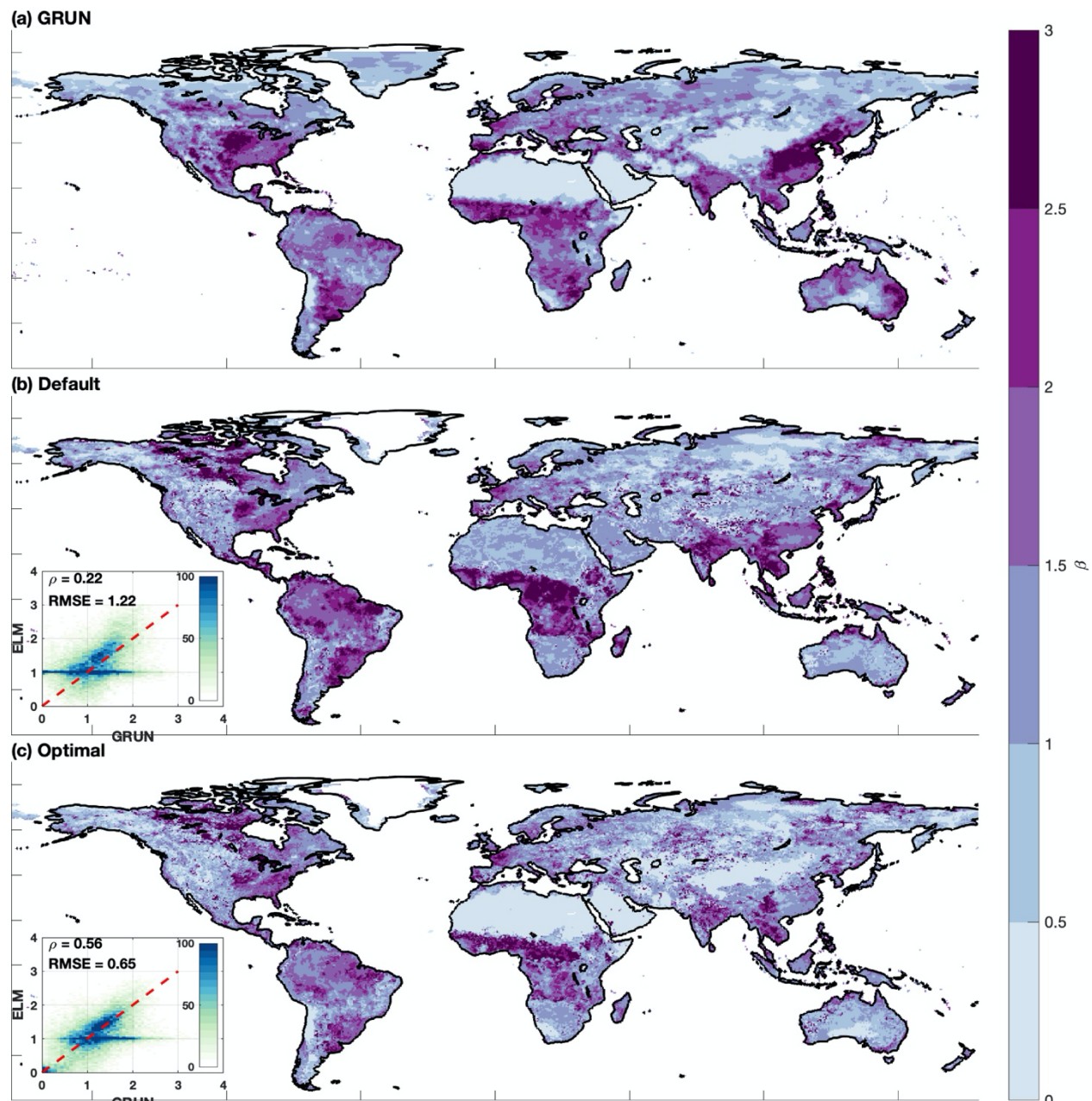

**Figure 9.** Sensitivity of runoff to precipitation ($\beta$) estimated from (a). GRUN runoff dataset, (b). ELM simulation with default parameter, and (c) ELM simulation with optimal parameter. The inserts show the scatter plots with density for cell-to-cell comparison of $\beta$ between GRUN and ELM simulations.

According to the evaluation with the ILAMB package, ELM shows similar performance in simulating other variables (e.g., latent heat flux, sensible heat flux, ET, and TWSA) with optimal parameter values compared to use of default parameter values

(Table 2). However, both the default and optimal simulations fail to capture the spatial variation of TWSA with the spatial distribution score less than 0.05. This is because the coarse resolution (e.g., several hundred km) of GRACE product (Seyoum et al., 2019) cannot resolve the spatial variability of TWSA for our model resolution.

**Table 2.** ILAMB benchmark scores for latent heat flux, sensible heat flux, evapotranspiration, and terrestrial water storage anomaly with default and optimized parameters in ELM. Description of each score metric can be found in http://redwood.ess.uci.edu/CMIP6_bnchmrk1_9_8/.

| Variable | Data source | Parameter | Bias Score | RMSE Score | Seasonal Cycle Score | Spatial Distribution Score | Overall Score |
|---|---|---|---|---|---|---|---|
| Latent Heat Flux | FLUXCOM | Default | 0.740 | 0.680 | 0.910 | 0.993 | 0.800 |
| | | Optimal | 0.730 | 0.677 | 0.909 | 0.992 | 0.797 |
| Sensible Heat Flux | FLUXCOM | Default | 0.682 | 0.643 | 0.932 | 0.940 | 0.768 |
| | | Optimal | 0.680 | 0.636 | 0.932 | 0.933 | 0.763 |
| ET | GLEAM3.3 | Default | 0.714 | 0.675 | 0.870 | 0.971 | 0.781 |
| | | Optimal | 0.705 | 0.672 | 0.873 | 0.967 | 0.778 |
| TWSA | GRACE | Default | 0.901 | 0.554 | 0.818 | 0.003 | 0.566 |
| | | Optimal | 0.900 | 0.545 | 0.817 | 0.004 | 0.562 |

**5.7 Parametric uncertainty**

The parameter priors listed in Table 1 result in significant uncertainties in the total runoff, with global average annual runoff for 1991-2010 varying from 30,999 - 76,496 [$km^3/yr$] (Figure 10a). After parameter inference, the uncertainty of the runoff constructed using simulations with parameter posteriors is constrained to 35,389 - 49,741 [$km^3/yr$]. The constrained

annual runoff uncertainty captures the reference runoff (38,443 [$km^3/yr$]) and is consistent with previous global runoff studies (Schellekens et al., 2017; Rodell et al., 2015; Clark et al., 2015; Haddeland et al., 2011). The simulation with the optimal parameter values yields an averaged global annual runoff of 42,156 [$km^3/yr$], overestimating the reference runoff by 9.6%. The overestimation is mainly from Amazon, Asia, and Eastern Europe (Figure 10b), and Ghiggi et al. (2019) reported a similar spatial bias pattern between global hydrological model simulations in ISIMP2a. The simulation with the default parameter

values shows smaller biases in terms of annual runoff magnitude as compared to the reference runoff data, with an overestimation of 5.3% on average. However, the smaller biases of annual runoff with the default parameters are because of cancelling out of the monthly errors to some extent. For example, the default parameters tend to overestimate the runoff during

the wet periods but underestimate the runoff during the dry periods in Amazon basin (Figure S10a). While the default simulation shows higher RMSE and lower NSE at monthly scale, it yields smaller biases at annual scale than the optimal 460 parameter (Figure S10b). Therefore, the calibrated simulation shows better performance in capturing the spatiotemporal variability (higher NSE and KGE in Figure 8b and e), but it doesn't lead to a reduced bias at annual scale. We further acknowledge that 200 simulations with 11 random parameters may not be sufficient to capture the full variations of simulated runoff.

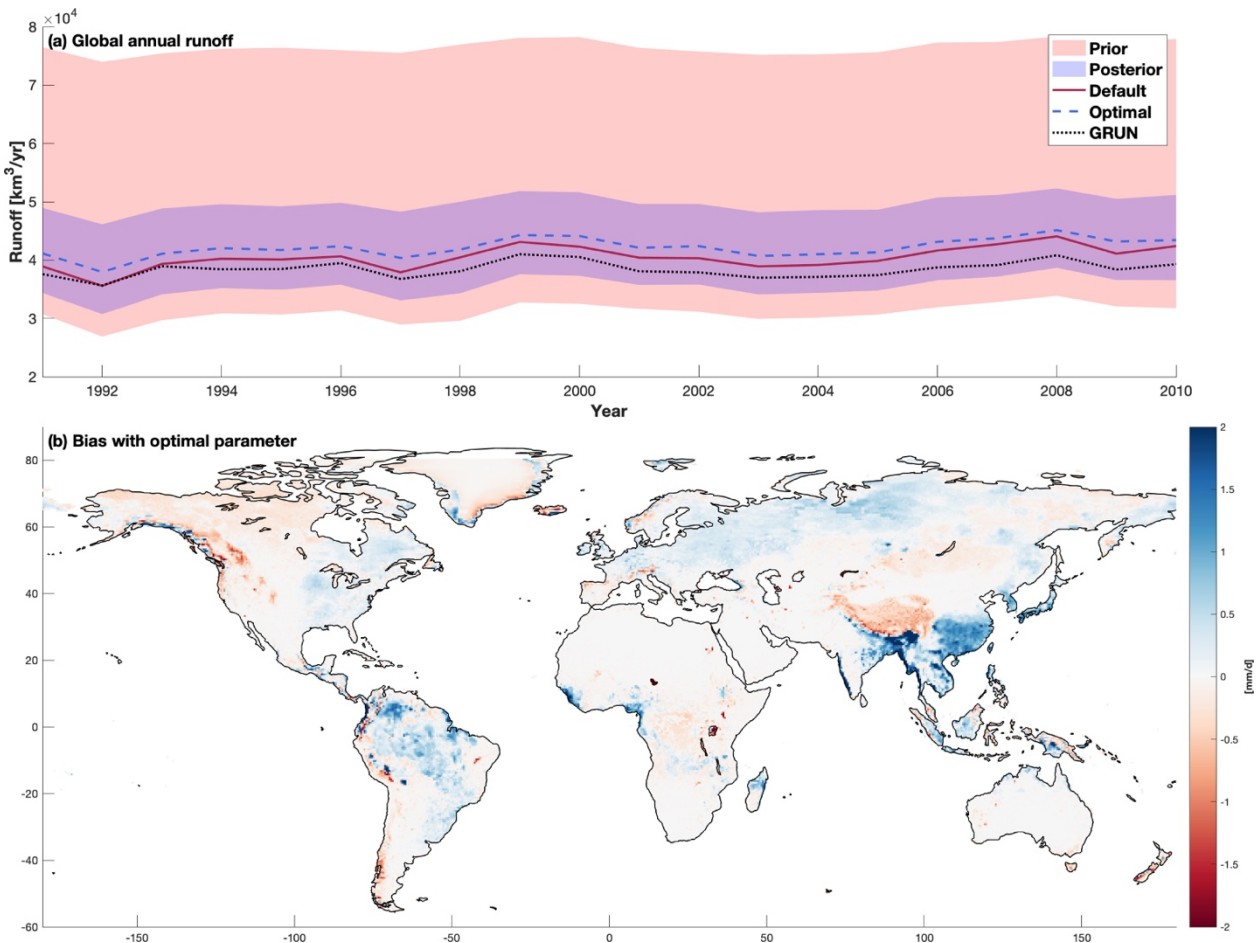

**Figure 10.** (a). Annual global runoff from default ELM simulation, optimal ELM simulation, and GRUN runoff dataset for the simulation period (1991-2010). The red and blue shade areas represent the uncertainties constructed from the simulations with parameter sampled on priors and posteriors, respectively. Subplot (b) shows the absolute difference of annual average 470 runoff between ELM simulation with optimal parameter and GRUN runoff data.

The runoff uncertainties associated with parameters are constrained significantly with the parameter posteriors at basin scale as well (Figure 11). Noticeably, the posterior uncertainty of annual runoff is larger over the equatorial regions (e.g., Parana, Amazon, Godavari, Congo) than other regions. The simulation with optimal parameter values yields larger overestimation of total runoff compared with the simulation using the default parameter values for the selected basins, except Mississippi, Godavari, and Loire basin (Table S1). The reason for the overestimations is that the optimal parameters are determined by maximizing NSE at monthly scale, which cannot ensure the annual runoff to be appropriately constrained. There exist significant discrepancies between simulations and GRUN for basins located at high latitudes (e.g., Mackenzie, Volga, Ob, Yenisey, and Lena) even when the posterior uncertainties are considered (Figure 11), highlighting the importance of snow-melting processes in snow-dominated regions. However, the large difference between ELM and the reference runoff in Yangtze river basin may be caused by the bias of the reference runoff since previous study reported annual discharge to be around 900 $[km^3/yr]$ (Yang et al., 2015).

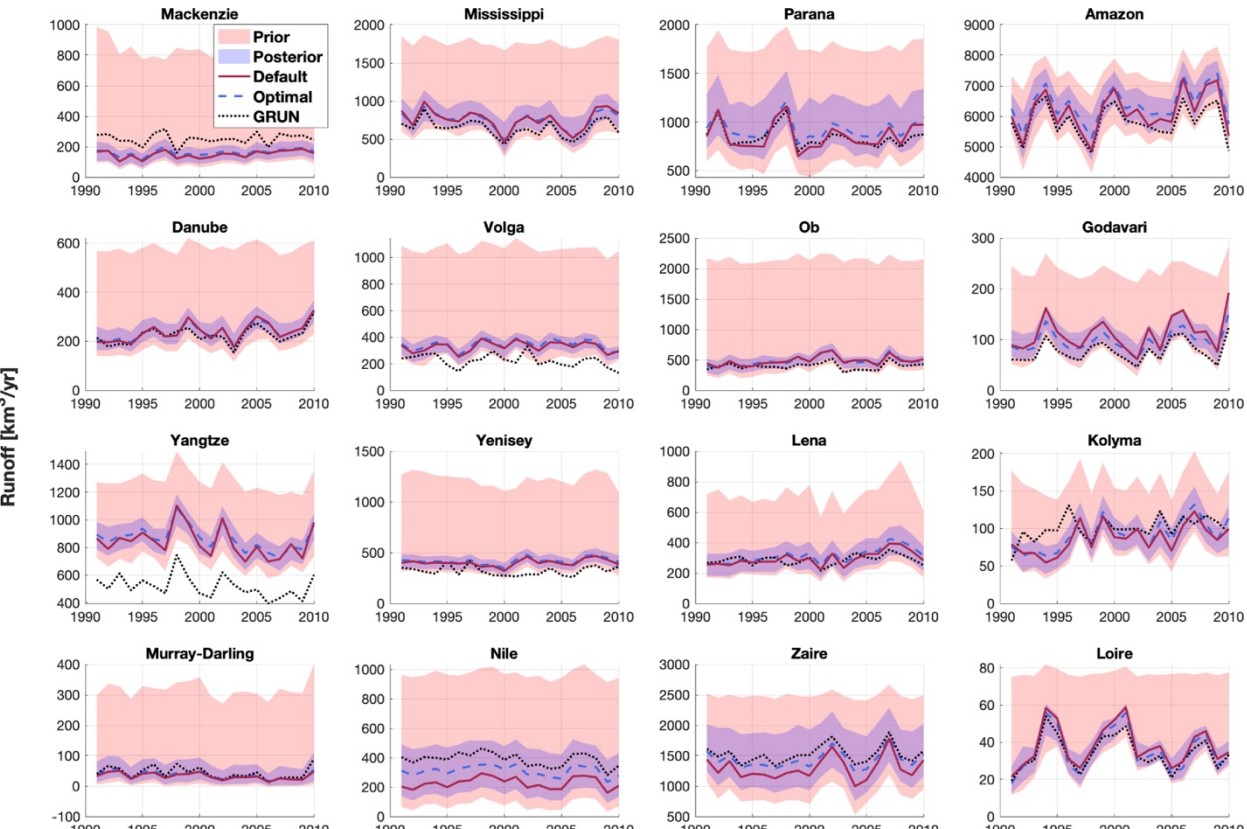

**Figure 11.** Annual runoff at basin scale from default ELM simulation, optimal ELM simulation, and GRUN runoff dataset for the simulation period (1991-2010). The red and blue shade areas represent the uncertainties constructed from the simulations with parameter sampled on priors and posteriors, respectively.

Despite being constrained by the parameter inference process, the parametric uncertainty of ELM-simulated annual runoff is considerable. Specifically, the posterior uncertainty of global runoff simulated by ELM is comparable to that of the multi-model ensemble constructed with the 13 global hydrological models from ISIMIP2a (Figure 12a). The parametric uncertainty affects not only the magnitude of global runoff but also the trend for the simulation period, during which a rapid increase of temperature has occurred (Figure S11a). The Sen's slope (Sen, 1968) for the reference runoff data is found to be 54.7 $[km^3/yr]$, but this increasing trend is not significant according to the Mann-Kendall test (Figure 12b). Other studies also reported no significant changes in the global runoff with observed streamflow data (Alkama et al., 2013; Dai et al., 2009; Milliman et al., 2008; Alkama et al., 2011). However, the default and calibrated ELM simulations yielded the Sen's slope to be 188.9 $[km^3/yr]$ and 133.8 $[km^3/yr]$ , respectively. Although the Sen's slope is reduced with the optimal parameters, the increasing trend remains significant. Likewise, all the other global hydrological models of ISIMIP2a exhibit significant increasing trend in the annual runoff, with the Sens' slope varying from 93 $[km^3/yr]$ to 272 $[km^3/yr]$ (Figure 12b). Considering the GRUN dataset and all model simulations are forced by the same atmosphere forcing (i.e., GSWP3), the differences of the global runoff trends can be attributed to the model structural/parametric uncertainty. We note that there exists a significant trend in GSWP3 precipitation at global scale, with an increase of 246.1 $[km^3/yr]$ during the simulation period (Figure S11b). But it remains unclear how the runoff responds to the increase of precipitation at global scale because the concurrent increased temperature (Figure S11a) leads to more ET, which can potentially balance the increased precipitation to some extent. The inconsistency of the global runoff trend between the model simulations and observation-based data can be caused by uncertainties of different sources. For example, the accuracy of GRUN is limited by the coverage of the streamflow gauges, as over half of the global areas are ungagged (Alkama et al., 2013). The model parametric uncertainty is significant, as ELM simulations with parameters posteriors show a wide range of annual runoff trend, from no trend to significant increasing trend (Figure 12b). This highlights the necessities of including parametric uncertainty in future runoff projections since runoff trend is not well constrained even if the model performance in the control period is improved.

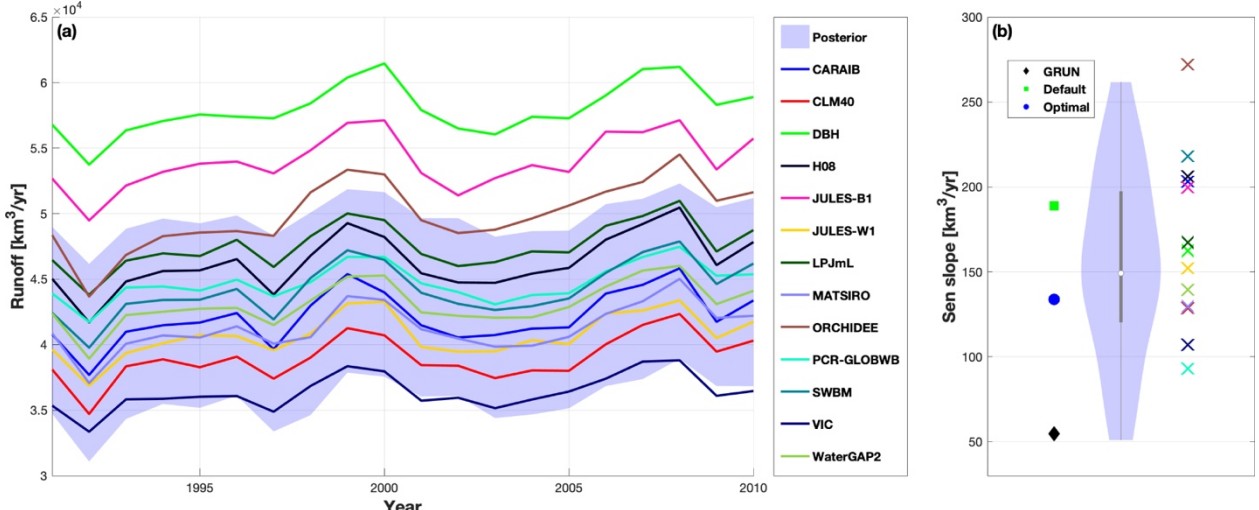

**Figure 12.** (a) Annual global runoff from 13 global hydrological models participated in ISIMIP2a, and ELM simulated runoff uncertainty constructed using simulations with parameter posteriors. (b) Sen's slope for the global annual runoff for the GRUN runoff dataset and simulations. The voilinplot (Hintze and Nelson, 1998) are generated with Sen's slope of ELM simulations with parameter posteriors, and the white point is the median values and the grey line represents range of the 25% - 75% percentile. The matlab fucntion of Bechtold (2016) was used to create the voilinplot. The cross signs are the Sen's slopes estimated from the ISIMIP2a model simulations.

## 6 Limitations

We note there can be other better choices of priors for the parameter whose range covers several orders of magnitude. For example, sampling $q_{drai,max}$ on a uniform distribution (e.g., $U[10^{-6}, 10^{-1}]$ results in fewer prior samples with values less than $10^{-2}$. A log-transformed uniform distribution can be a good alternative to guarantee enough samples over each range of the desired values. Using a log-transformed uniform distribution for $q_{drai,max}$ prior doesn't impact our results significantly because the simulated runoff is not sensitive to $q_{drai,max}$ (Figure S4), and more samples over smaller values of $q_{drai,max}$ will not lead to more variation in runoff. However, careful selection of prior distributions can be important for sensitive parameters in future application of surrogate-assisted calibration framework.

By using RMSE instead of the simulated runoff as the QoI, only one PCE-based surrogate model is constructed for each grid cell to represent the ELM performance of simulating monthly runoff time series. Although selecting RMSE as QoI significantly reduces the computational burden of surrogates' construction and parameter inference, the corresponding surrogates cannot be used to estimate posterior uncertainty of physical model outputs. For example, we still need to run ELM simulations after the parameter inference to construct the runoff posterior uncertainty (Sec 3.5). Additionally, the objective of this study is to minimize RMSE at monthly scale, hence an improved model performance at annual scale is not guaranteed.

Including both monthly and annual performance metrics in objective function may balance the performance at different temporal scales. However, only one objective is accepted in the uncertainty quantification framed used in this study.

We further acknowledge the poor performance of PCE-based surrogate model in capturing the ELM-simulated runoff over extremely arid regions (Figure 2 and 3). This can be attributed to the limitation of polynomial-based surrogate models in capturing highly non-smooth or strongly nonlinear relationships. Machine learning algorithms (Dagon et al., 2020) and deep neural network (Tsai et al., 2021) are alternative techniques for surrogate modelling, which are better at capturing non-smooth or nonlinear functions, but future research is needed to investigate the capability.

The calibrated parameters have a significant impact on baseflow index, which is the ratio between subsurface runoff and total runoff. For example, the baseflow of Amazon basin with default and optimal parameters are 0.53 and 0.70, respectively (Figure S12). Mortatti et al. (1997) reported the baseflow index of Amazon basin to be 0.70 with isotopic tracer method, which is consistent with our simulation with optimal parameter values. However, accurate separation of surface runoff and subsurface runoff over other regions is not guaranteed, though the total runoff has been calibrated to match with the reference runoff dataset. The global baseflow index dataset of Beck et al. (2013) that derived from observed streamflow

provides us the benchmark for evaluating the baseflow index simulated in ELM. Constraining the baseflow index during the ELM validation and calibration study will be investigated in the future.

      We further note that uncertainty in the reference runoff data of GRUN used in the parameter inference is inevitable. While Ghiggi et al. (2019) found that GRUN outperformed other global hydrological models and multi-model ensemble, lower

accuracy over mountainous regions due to the coarse resolution has been reported. Additionally, the irrigation and water management impacts on streamflow was included for some regions during the training process of GRUN (Ghiggi et al., 2019), but irrigation and water management are not active in ELM configuration used in this study. This inconsistency may explain the significant overestimation of ELM simulated runoff compared to GRUN for certain regions, for example, Yangtze River basin (Figure 10).

Another limitation of this study is that the snow melting processes were not calibrated. A poor representation of snow melting process can result in poor skill of runoff generation in snow-dominant regions, where snowmelt is an important contribution to runoff (Jenicek and Ledvinka, 2020). This could explain the low performance (i.e., negative NSE) of calibrated ELM over the Northern Hemisphere high latitudes and mountainous regions. However, including parameterizations of snow processes such as snow albedo, solar absorption, and snow aging (Lawrence et al., 2011) can introduce more uncertain

parameters, which will make calibration more challenging (Huang et al., 2013). In the future, a dedicated calibration on the snow melting process is needed to improve the runoff generation in snow-dominated regions.

**7 Conclusion**

      In this study, we applied an UQ framework to calibrate the runoff generation relevant parameters in the ELM-v1 using an observation-based runoff dataset as benchmark. The parameters with higher sensitivity are identified through the

sensitivity analysis with the PCE-based surrogate models. While different sensitivity patterns are found for different regions, 81.3% of the global cells show significant sensitivities to three or fewer parameters of the 11 selected parameters. The results of our sensitivity analysis are consistent with those of previous studies over the US continent (Huang et al., 2013; Sun et al., 2013), with runoff showing the largest sensitivity to the subsurface runoff parameter. The Bayesian posterior distribution of the highly sensitive parameters at each grid cells is estimated with MCMC simulations, using the surrogate model to construct the likelihood function. Additional ELM simulations with parameter samples from the posterior run to estimate the optimal parameter values and construct the parametric uncertainty for the simulated runoff. While the optimal parameter values improve the model performance of runoff significantly, the parametric uncertainty is comparable to the uncertainty in a multi-model ensemble in ISIMIP2a, which is appreciable. Furthermore, the parameters are found to impact the annual global runoff trend for our simulation period. Specifically, the simulations with parameter posteriors show a wide range of the annual runoff trends at global scale, from no trend to significant increasing trend. In summary, parameter calibration is necessary to improve model performance and parametric uncertainties should be considered for comprehensive analysis of runoff and its projections.

**Code and Data Availability**

The current version of ELM is available from E3SM project (https://github.com/E3SM-Project/E3SM/releases/tag/v1.1.0). The UQTk code and documentation are available from https://www.sandia.gov/uqtoolkit/. The exact version of ELM, exact version of UQTk source code, and scripts to produce the plots in this study is archived on Zenodo (https://doi.org/10.5281/zenodo.5815500). Matlab version R2019b Update 4 was used to run the processing and plotting scripts. ILAMB version 2 was used in this study, and the package can be accessed at 10.18139/ILAMB.v002.00/1251621. The domain file and surface data file that used to run ELMv1, and processed ISIMP2a runoff data are archived on Zenodo (https://doi.org/10.5281/zenodo.5815730). The GRUN runoff dataset was downloaded from https://doi.org/10.6084/m9.figshare.9228176.

**Author Contribution**

DX and GB designed the study. DX run the simulations, performed the analysis, visualized the results, and prepared the first draft of manuscript. GB mentored DX through this study. KS helped with the Uncertainty Quantification methodology. CL investigated the results. All authors contributed to the discussion and review of the results and manuscript writing.

**Competing interests**

The authors declare that they have no conflict of interest.

## Acknowledgments

This work was supported by the Earth System Model Development program area of the U.S. Department of Energy, Office of Science, Office of Biological and Environmental Research as part of the multi-program, collaborative integrated Coastal Modeling (ICoM) project. The Pacific Northwest National Laboratory is operated by Battelle for the U.S. Department of Energy under Contract DE-AC05-76RLO1830. CL was supported through Next Generation Ecosystem Experiments-Tropics, funded by the U.S. Department of Energy, Office of Science, Office of Biological and Environmental Research at Pacific Northwest National Laboratory. Sandia National Laboratories is a multi-mission laboratory managed and operated by National Technology and Engineering Solutions of Sandia, LLC., a wholly owned subsidiary of Honeywell International, Inc., for the U.S. Department of Energy's National Nuclear Security Administration under contract DE-NA-0003525.

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
