# Peer review of "Using a Surrogate-Assisted Bayesian Framework to Calibrate the Runoff Generation Scheme in E3SM Land Model V1"

_Geoscientific Model Development, 2021_

## Author Response (AR1)

**Reviewer 1**

This work presents a surrogate-assisted framework for calibrating runoff relevant parameters in global-scale Earth System Models (ESMs). The large computation burden arisen from repeated simulations in calibration is alleviated by building fast-to-run PCE-based surrogate models of ESMs. It is concluded that the calibrated model obtains an improved performance compared to the one with default parameter values. In summary, the manuscript is generally well-written and may be eventually accepted after addressing the following comments:

Response: We thank the reviewer for the constructive comments that have helped us improve the manuscript. Please find our point-by-point responses to the comments provided below, and the corresponding modifications in the revised manuscript.

-The title should be revised. In my opinion, uncertain quantification is different from calibration. How can one use a UQ framework to calibrate models? How about "Using a surrogate-assisted Bayesian framework to ..."?

Response: As suggested, we modified the title in the revised manuscript.

-In table 1, the prior for q_{drai,max} is U(1e-6,1e-1). Why not use a logarithmic transformation for it? Otherwise, much more prior samples will be drawn from, e.g., (1e-2, 1e-1).

Response: We appreciate the reviewer for bringing our attention to the sampling issue. The use of a uniform prior distribution does result in fewer samples of $q_{drai,max}$ near the lower bound of the prior distribution, and using a logarithmic transformed uniform distribution suggested by the reviewer results in samples that better represent the entire range of the prior distribution. However, additional tests show that use of a uniform distribution does not alter the finding of our study. Our framework consists of following three steps: (1) parameter sensitivity analysis using the surrogate model, (2) inference of optimal values for the most sensitive parameters identified in step 1, and (3) model calibration using the posteriors of the inferred parameters. In step 1, the runoff performance was found to be insensitive to $q_{drai,max}$ compared to other parameters (Figure S4), thus it was not used in steps 2 and 3.

We have now tested the impacts of different prior distributions on the sensitivity of $q_{drai,max}$, which is used to calculate subsurface runoff, $R_{drai}$, using the following equation:

$$R_{drai} = q_{drai,max} \exp(-f_{drai} z_\nabla)$$

where $q_{drai,max}$ is the maximum drainage rate, $f_{drai}$ is the decay factor, and $z_\nabla$ is the water table depth. We tested the differences of variation of $R_{drai}$ caused by using two different sampling methods for $q_{drai,max}$: uniform distribution (used in the main text), and log-transformed uniform distribution (suggested by the reviewer).
Using 200 samples (number of training and validation simulations for surrogate construction), the standard deviation of $R_{drai}$ is always larger for uniform distribution compared to log-transformed uniform distributions for different $f_{drain}$ (different subplots in Figure 1) and water table depth (X-Axis of subplots in Figure 1). The global sensitivity analysis in this study used

Sobol index (variance based), therefore, using the log-transformed uniform distribution will lead to smaller Sobol index. Since $q_{drai,max}$ remains an insensitive parameter even with log-transformed uniform distribution, the conclusion of our study remains unchanged.

Although we think our results will not be affected by using another prior, we think this is a good point to highlight in the main text because we agree the selection of prior distribution can be important in other applications (line 528-line534).

[Figure]

Figure 1. Comparison of standard deviation ($\sigma$) of subsurface runoff ($R_{drai}$) with samples from uniform distribution (blue solid line), and samples from log-transformed uniform distribution (red dashed line). Note, the Y-Axis is log transformed.

-Line 193 and Eq. (17), the authors should clearly present how they determine the values of sigma.

Response: The objective of this study is to identify ELM parameters that minimize the RMSE between the ELM-simulated runoff and the GRUN runoff observation. Thus, the sigma is estimated as the standard deviation of the RMSE computed from all the training simulations and

the reference runoff (i.e., GRUN). We have clarified the methodology for computing sigma in line 227-line228.

-In section 3.3, which criterion (e.g., the Gelman-Rubin R statistic [Gelman et al., 1995]) is used here to check the convergence of MCMC sampling? From Fig. 6 it can be seen that the posterior ranges are still relatively large, which gives the feeling that the MCMC chain has not totally converged.

Response: Thanks for providing the reference for assessing the convergence of MCMC chain. We evaluated the Gelman-Rubin R statistic for five MCMC chains using 10,000 samples for the posterior distributions of $f_{drain}$, $f_{over}$, and $\psi_s$ (Figure 6 (a-c)). The Gelman-Rubin R statistic for $f_{drain}$, $f_{over}$, and $\psi_s$ are 1.002, 1.004, 1.003, respectively. The MCMC chain did converge as the statistic for all three parameters is close to 1.0. We have added a discussion about the Gelman-Rubin R statistic in the revised manuscript on line 365 - line367.

-In section 5.2, the poor performance of PCE surrogate models in arid regions probably because of PCE's inability to approximate highly nonlinear functions (a well-known limitation of PCE) or/and the low signal-to-noise ratio in these regions. The authors should elaborate these to provide more informative results to readers. An alternative surrogate method for approximating highly nonlinear function is the deep neural networks.

Response: We have now added a discussion about the inability of PCE to capture the model behavior in extremely dry regions due to the high nonlinearity and non-smooth behaviors of simulated runoff as a limitation of our study. We added discussions of alternative options as the reviewer mentioned, such as deep neural networks, or other machine learning methods (line 535-line539).

-In Figs. 10a and 11, the simulated runoff time series with default parameter values could be even closer to the reference GRUN time series than the calibrated ones, giving the feeling that the calibration is not that satisfactory. Please further explain.

Response: In the original submission (line 440 – line 446), we discussed the potential reasons for a higher bias in ELM simulation at the annual scale using optimal parameters. The objective of our model calibration is to improve the model performance at monthly scale, therefore the bias at annual scale with the optimized parameters may not be smaller as compared to the bias with the default parameters. We have now added text on line 460 – line 463 and a figure in supplementary material (Figure S9b) to acknowledge that the default parameters lead to a lower bias compared to optimized parameters at annual scales, which yield lower RMSE and higher NSE at monthly scale.

-Line 533, 'are estimated', Line 534, 'are run'?

Response: Thanks! We corrected this typo.

**Reviewer 2**

The manuscript performed a per-grid calibration of the E3SM ELM model against a global monthly runoff dataset. The calibration was enabled by developing surrogate models for each grid of the ELM using Polynomial Chaos Expansion to mimic the response surface, which was chosen to be the root mean square error of monthly runoff for each grid. Subsequent analyses examined the spatial distribution of calibrated parameters with higher sensitivity and parametric uncertainty effects on simulated runoff. The paper is well organized, clearly written, and deals with an important topic of calibrating ELM and similar models. However, I have some concerns regarding the accuracy of surrogate and its effect on calibration, as detailed below.

Response: We thank the reviewer for the constructive comments that have helped us improve the manuscript. Please find our point-by-point responses to the comments provided below, and the corresponding modifications in the revised manuscript.

Line 15: "The main methodological advance in this work is the construction of surrogates for the error metric between the ELM and the benchmark data". But this is not entirely new as using surrogate in this manner has been done previously, e.g. Wang et al. (2014); Razavi et al. (2012) and references therein.

Response: Thank you for providing the references that leverage surrogate modelling in model calibration. We agree with the reviewer that calibration and uncertainty quantification using surrogate model method have been applied in hydrology previously. We provided a few references in the original manuscript in line 82-line 85 and have now included additional references provided here in the revised manuscript to provide the broader context for our work.

The advance of our work is **the selection of Quantity of Interest (QoI)**, rather than applying surrogate model for calibration. A typical study with monthly runoff as the QoI that uses 20 years of data requires constructing 240 (= 20 years x 12 months) surrogates for each grid cell using the PCE method. It is not computationally tractable to construct PCE surrogates for a global domain containing 70,302 grid cells, which result in $70,302 \times 240 = 16,872,480$ surrogates. The computational cost of parameter inference will be even higher as it requires tens of thousands of simulations with all the surrogates to generate MCMC chains. Therefore, in this study we selected ELM-simulated runoff RMSE as the QoI, which results in the construction of only one PCE-based surrogate model for each grid cell to represent the ELM performance of simulating monthly runoff time series. RMSE is commonly used as objective function for parameter inference process. The novelty of our study is the construction of surrogate for RMSE instead of runoff, which can be directly used in later MCMC simulation (see Eq 21, which is equivalent to Eq 18). The selection of RMSE as QoI has been discussed in Sec 3.4, we further clarify the selection of RMSE as QoI as the novelty of our work in the abstract (line 15 – line 16) and on line 213 – line 214, and line 564 – line 566 of the revised manuscript.

Line 111: what's the difference between surface runoff and surface water runoff?

Response: Surface runoff represents the saturation excess runoff (i.e., Dunne runoff), and surface water runoff is the water drainage from the wetland. We have clarified the definition of these two types of runoffs in line 114 – line 115.

Line 195 - to reduce the log likelihood to least-squares regression, further assumption is needed, which might include constant and known sigma. Please verify.

Response: The assumption of least-squares regression is that the error between the model simulations and reference data follows the normal distribution with a vanishing mean (line 196). The corresponding $\sigma$ is estimated from the data, such as the standard deviation of all the RMSE between the training simulations and GRUN runoff data at each grid cell. We have added text to clarify the estimation of sigma on line 227 – line 228 of the revised manuscript.

Line 197 - I am not sure whether 1,000 samples are sufficient for burn-in, since MCMC often requires a large number (e.g., tens of thousands) of samples to converge. Including some convergence check statistics or plots in supplementary material would be helpful. Also, what is the MCMC algorithm being used here? Please include a reference for reproducibility.

Response: As also pointed out by reviewer 1, we have added an evaluation of the metric of Gelman-Rubin R statistic with five MCMC chains using 10,000 samples from the posterior distributions of $f_{drain}$, $f_{over}$, and $\psi_s$ (Figure 6 (a-c)). The Gelman-Rubin R statistic for $f_{drain}$, $f_{over}$, and $\psi_s$ are 1.002, 1.004, 1.003, respectively. The MCMC chain did converge as the statistic for all three parameters is close to 1.0. We have added a discussion about the Gelman-Rubin R statistic in the revised manuscript on line 365 – line 367.

We used adaptive MCMC algorithm which updates proposal covariance on-the-fly, according to the current chain history. We included the reference (Haario et al., 2001) of this particular flavor of MCMC algorithm in the revised manuscript on line 204 – line 205.

Fig. 1& 2 - there's some discrepancy between RMSE given by the surrogate and by ELM. Studies have shown that even small surrogate error can lead to large deviation of the inferred parameter posterior from the "true" posterior (Laloy and Jacques, 2019). I realize that it is not possible to calibrate ELM at global scale, but it seems possible to perform some quick test to validate the surrogate modeling approach. For example: for a few grids compare the posterior obtained using PCE and using ELM; In Section 3.5, step #4, compare the RMSE of ELM simulation with that of PCE.

Response: The optimal parameters inferred from the surrogate models may not yield minimum RMSE for ELM-simulated runoff at monthly scale. Thus, the surrogate model was first used to find the most sensitive parameters and the corresponding posterior distributions, which were more significantly constrained than the priors. Next, we perform additional 100 ELM simulations to find the optimal ELM parameters and the runoff uncertainties. So, the surrogate models were used to identify the most sensitive parameter and estimate the corresponding posterior of the sensitive parameter. Finally, the optimal parameter and runoff posterior were estimated based on ELM simulations instead of surrogate models.

We cited (Laloy and Jacques, 2019) in line 239 to highlight the issue of surrogate error.

We have now clarified that the additional 100 ELM simulations were used to find the optimal parameters and construct runoff uncertainty on line 240 and line 243of the revised manuscript.

Line 355: If I understand correctly, 10,000 is the number of runs of the surrogate. It is not necessarily the case if ELM is run, because the convergence rate may be different given the surrogate error (Razavi et al., 2012).

Response: The PDFs of RMSE (Figure 6) are indeed generated using 10,000 runs with surrogate models. This figure is to show that fewer samples on parameter posterior are needed to find the optimal parameter corresponding to the minimum RMSE than sampling on the priors. The sensitivity analysis identifies the most sensitive parameters in each grid cell, and the MCMC simulation is used to obtain the posterior of the most sensitive parameters with a significantly constrained range. Then, a set of 100 ELM calibration simulations were performed with parameter sampled from the posteriors of the three most sensitive parameters to find the optimal parameters and construct the runoff parametric uncertainty.

We have now clarified the description of the surrogate simulations on line 368 and line 370 in the revised manuscript.

Fig. 11 - it seems that the same period of 1997-2010 is used to calibrate the model and validate the optimal parameters. Is data available after 2010 for validation, so that validation data is independent from calibration data?

Response: Yes, the GSWP3 forcing is available after 2010, such as 2011-2014. We extend our simulation to 2013 (unfortunately, the GSWP3 forcing of 2014 is problematic in our system), and evaluation of the simulation with calibrated parameter and default parameter. The result is added in the supplementary materials (Figure S8) and discussed in line 414 – line 416 in the revised manuscript.

Some paragraphs are indented, some are not.

Response: We have fixed the indentation in the revised manuscript.

**Reference**

Laloy, E., & Jacques, D. (2019). Emulation of CPU-demanding reactive transport models: a comparison of Gaussian processes, polynomial chaos expansion, and deep neural networks. Computational Geosciences, 23(5), 1193-1215.

Wang, C., Duan, Q., Gong, W., Ye, A., Di, Z., & Miao, C. (2014). An evaluation of adaptive surrogate modeling based optimization with two benchmark problems. Environmental Modelling & Software, 60, 167-179.

Haario, H., Saksman, E., & Tamminen, J. (2001). An Adaptive Metropolis Algorithm. Bernoulli, 7(2), 223–242. https://doi.org/10.2307/3318737

---

## Author Response (AR2)

General response: We appreciate reviewers' efforts for reviewing our revised manuscript. We revised the manuscript based on the following comments to further strengthen our manuscript. Please find our point-by-point responses in the following.

Reviewer #1
The authors have addressed most of my comments. I have some additional comments after reading the responses and revised manuscript.

-In the first-round review, a concern was about the better performance of the model with the default parameters than that with the optimal parameters. In the revised manuscript, the authors added an explanation in L 460-462 but dismissed quite quickly. Since the aim of model calibration is to obtain more reliable predictions (the focus of this work), the revision does not totally address the concern. The authors should elaborate the *potential deep causes* here.

Response: In this revision, we have now explicitly highlighted in the introduction section (Line 96) that the time scale for the calibration of ELM-simulated runoff is monthly. The optimal parameters thus obtained for calibration of monthly ELM-simulated runoff cannot guarantee an improved model performance at annual scale with respect to the default parameters. We also added explanations in the revised manuscript at Line 546 – Line 549.

-Please show the convergence curve of Gelman-Rubin R statistic in the manuscript.

Response: We added the convergence curve of Gelman-Rubin R statistic in the supplementary material Figure S6.

-The authors proposed to build a surrogate model for the RMSE metric to avoid constructing a surrogate for each grid. This is a good idea but a discussion on the potential limitation of this strategy would be helpful.

Response: We added two limitations in the revised manuscript. First, the surrogates of RMSE cannot be used to construct runoff uncertainty, therefore, ELM simulations are still needed. Another limitation is using RMSE at monthly scale as objective cannot guarantee the performance at annual scale. Please find details in Line 542 to Line 549.

Reviewer #2
I appreciate the authors taking the effort to address my comments, particularly related to adding a new period of simulation to test the calibrated model. While the revision has improved the manuscript, it does not address some of my comments, re-iterated below. It seems that some necessary (and relatively minor) changes are needed before the manuscript can be accepted.

- The novelty is still unclear to me. The authors clarified in the revision that "The selection of RMSE as QoI in constructing surrogate models is a novelty of this work, which can significantly reduce the computational burden of surrogates' construction and parameter inference." (line 213-214 in the manuscript with tracked changes). As pointed out in the previous review comments, this (using RMSE as QoI) is not new (e.g. Wang et al. (2014); Razavi et al. (2012) and references therein). Developing a surrogate model for a performance measure (RMSE in this case) and then

optimizing it is actually the focus of Razavi et al. (2012) paper cited in the manuscript, and these methods have had a good number of hydrologic applications in the last decades. Therefore, I would suggest the authors highlighting their contribution in the analysis results such as runoff patterns before/after calibration, rather than the surrogate modeling method itself.

Response: We reworded the sentences that highlight the selection of RMSE as QoI is the novelty in the revised manuscript as listed in the following.

In abstract, we modified "The main methodological advance is this work is the selection of error metric between the ELM simulations and the benchmark data is selected to construct the surrogates, which facilitates efficient calibration and avoids the more conventional, but challenging, construction of high-dimensional surrogates for the ELM simulated runoff." to "Error metric between the ELM simulations and the benchmark data is selected to construct the surrogates, which facilitates efficient calibration and avoids the more conventional, but challenging, construction of high-dimensional surrogates for the ELM simulated runoff."

In the method section, we modified "The selection of RMSE as QoI in constructing surrogate models is a novelty of this work, which significantly reduce the computational burden of surrogates' construction and parameter inference." to "The selection of RMSE as QoI in constructing surrogate models significantly reduce the computational burden of surrogates' construction and parameter inference."

- Thank you for adding information regarding the convergence of the MCMC chains. Please further clarify at what stage the reported Gelman-Rubin R statistics were evaluated, i.e., were they calculated at the 1,000th iteration (the end of burn-in), or the 10,000th sample. Only samples after convergence can be retained to form the posterior distribution, so the R statistics at the 1,000th iteration is needed here.

Response: The Gelman-Rubin R statistics were estimated with the samples after the burn-in period. We have now clarified it at Line 371.

- Likelihood function Eq. (21) - I don't follow the added text "where sigma is estimated as the standard deviation of RMSEs between simulated runoff and GRUN from all the training simulations because the objective is to minimize RMSE". If I understand correctly, sigma should refer to the standard deviation of the error (difference between GRUN runoff and simulated runoff), different from the text. For example, RMSE is always positive, but error can have both signs.

Response: Yes, the reviewer is right that $\sigma$ refers to the standard deviation of the error. In Equation (18), the error represents the difference between GRUN runoff and simulated runoff when runoff is the QoI. However, since we used the RMSE as QoI for constructing surrogate, Eq (18) is transformed to Eq (21) when evaluating likelihood in the parameter inference process:

$$\log L(\boldsymbol{y}|\boldsymbol{X}) = -\frac{N \cdot (0 - RMSE^{PC})^2}{2\sigma^2} - \frac{N}{2}\log(2\pi\sigma^2)$$

Here, we assume $\sigma$ refer to the standard deviation of difference between RMSE and 0 because RMSE is our QoI and 0 represents RMSE of observation. In other words, the objective is to find

the parameter with RMSE as close to 0 as possible. So, we estimated $\sigma$ as the standard deviation of RMSE between simulated runoff and GRUN from all the training simulations. In this revision, we modify Eq (21) to show RMSE = 0 is the objective, and clarify $\sigma$ has a different meaning than previous equation (Line 223 – Line 225).

- Following up on the other reviewer's comment - since the optimal parameters fit the monthly runoff better than default but not the annual, I suggest replotting Fig. 11 to show monthly runoff and discussing the calibration gains on capturing the monthly runoff patterns.

Response: Apart from calibrating the ELM-simulated runoff at monthly scale, the study also aims to quantify the parametric uncertainty of the simulated runoff and understand how the parametric uncertainty impacts annual runoff trend. Therefore, in Sec 5.7, we present analysis of the annual runoff to illustrate the parametric uncertainty and perform trend analysis. The Figure 11 shows the parametric uncertainty of the annual runoff at basin scale is significantly reduced after parameter inference. Additionally, plotting Figure 11 at monthly scale will make the figure very busy, thus affects the readability.

---

## Author Response (AR3)

Comment:
The reviewers looked through the revisions and were satisfied with most of the responses except the one regarding sigma in Eqn. (21). The equation and interpretation assume that RMSE should follow N(0,sigma^2). But the reviewer don't see how this can be derived from the assumption that errors follow N(0, sigma_e^2), given that RMSE is a nonlinear function of errors. Please respond before it can be finally accepted for publication.

Response:
The following shows the detailed derivation of Eq (21) from Eq (18).

$$\log L(\mathbf{y}|\mathbf{X}) = -\sum_{i=1}^{N} \frac{(y_i - \mathcal{M}_i(\mathbf{X}))^2}{2\sigma^2} - \frac{N}{2}\log(2\pi\sigma^2)$$

$$= -\frac{N \times \frac{1}{N}\sum_{i=1}^{N}(y_i - \mathcal{M}_i(\mathbf{X}))^2}{2\sigma^2} - \frac{N}{2}\log(2\pi\sigma^2)$$

$$= -\frac{N \times RMSE^2}{2\sigma^2} - \frac{N}{2}\log(2\pi\sigma^2)$$

$$= -\frac{N(0 - RMSE)^2}{2\sigma^2} - \frac{N}{2}\log(2\pi\sigma^2)$$

where $\sigma$ represents the standard deviation of the error between simulation and reference runoff based on the following assumption:

$$\epsilon_i \sim \mathcal{N}(0, \sigma^2), i = 1, 2, \ldots, N$$

Here subscript $i$ represents $ith$ month from 1991-2010, $N = 240$, and the underlying assumption is the error of simulated runoff has the same distribution for each month. It is not clear how to estimate $\sigma$ with runoff data because model exhibits different error distributions for different months. Specifically, Figure 1 shows that the $\sigma$ estimated with model simulations and observation for all the 240 months (i.e., 20 year time series) varies significantly. However, focusing on Eq (21), the error term can be further assumed as the difference between RMSE and 0 as RMSE is selected as QoI, and one can consider RMSE is the model simulation (i.e., $\mathcal{M}(\mathbf{X}) = RMSE$) and 0 is the observation (i.e., $y = 0$). So, we estimate $\sigma$ in Eq (21) using the standard deviation of difference between the simulated RMSEs and 0. In this way, we include runoff data from all months for estimating $\sigma$. We acknowledge there can be other approximations for $\sigma$, but our estimation yields reasonable posterior of parameters and runoff. It is beyond the scope of this study to investigate the impacts of $\sigma$ on the parameter inference. We added the following statement in Line 221 – Line 227 to clarify this issue.

"The standard deviation of error between model simulated runoff and observation exhibits significant monthly variation. To provide a reasonable value of $\sigma$, we further assume $\sigma$ in Eq (21) has a different meaning than that in Eq (18) by taking RMSE as model simulation, and 0 to as the target. Therefore, $\sigma$ is approximated as the standard deviation of the difference between 0 and RMSEs, where each RMSE was calculated using simulated runoff and observation for a given training simulation. Our estimation of $\sigma$ leads to a reasonable posterior (see Sec 5.4), though other methods can also be used to estimate $\sigma$. We acknowledge that the value of $\sigma$ may have an impact on the parameter posterior, but investigating the sensitivity of $\sigma$ on the posteriors is beyond the scope of this study."

[Figure]

**Figure 1.** Histogram of $\sigma$ estimated from error between simulated runoff and GRUN for each month during 1991-2010 at the example grid cell (e.g., Figure 6 in main text). The red line represents the $\sigma$ estimated with the standard deviation of RMSEs used in this study.